# POSTERIOR SAMPLING VIA LANGEVIN DYNAMICS BASED ON GENERATIVE PRIORS

## ABSTRACT

Posterior sampling in high-dimensional spaces using generative models holds significant promise for various applications, including but not limited to inverse problems and guided generation tasks. Despite many recent developments, generating diverse posterior samples remains a challenge, as existing methods require restarting the entire generative process for each new sample, making the procedure computationally expensive. In this work, we propose efficient posterior sampling by simulating Langevin dynamics in the noise space of a pre-trained generative model. By exploiting the mapping between the noise and data spaces which can be provided by distilled flows or consistency models, our method enables seamless exploration of the posterior without the need to re-run the full sampling chain, drastically reducing computational overhead. Theoretically, we prove a guarantee for the proposed noise-space Langevin dynamics to approximate the posterior, assuming that the generative model sufficiently approximates the prior distribution. Our framework is experimentally validated on image restoration tasks involving noisy linear and nonlinear forward operators applied to LSUN-Bedroom (256 x 256) and ImageNet (64 x 64) datasets. The results demonstrate that our approach generates high-fidelity samples with enhanced semantic diversity even under a limited number of function evaluations, offering superior efficiency and performance compared to existing diffusion-based posterior sampling techniques.

## 1 INTRODUCTION

Generative models that approximate complex data priors have been leveraged for a range of guided generation tasks in recent years (Dhariwal & Nichol, 2021; Chung et al., 2023). Early works focused on conditional synthesis using Generative Adversarial Networks (GANs) (Goodfellow et al., 2014; Mirza & Osindero, 2014; Brock et al., 2019; Karras et al., 2019; 2020). However, diffusion models have recently surpassed GANs as the state of the art in generative modeling (Ho et al., 2020; Song et al., 2021a), demonstrating superior performance in guided generation tasks (Dhariwal & Nichol, 2021; Choi et al., 2021; Ho & Salimans, 2021). Posterior sampling, as a guided generation framework, has garnered significant interest (Kawar et al., 2021; 2022; Chung et al., 2023), particularly for providing candidate solutions to noisy inverse problems.

Solving noisy inverse problems involves reconstructing an unknown signal $x$ from noisy measurements $y$, where the forward model is characterized by the measurement likelihood $p(y \mid x)$. The objective is to sample from the posterior distribution $p(x \mid y) \propto p(y \mid x)p(x)$. Such posteriors are often intractable in practical applications due to the complexity of the prior distribution $p(x)$. However, learned generative models that approximate complex data priors can enable approximate sampling from the posterior $p(x \mid y)$. Early approaches for solving inverse problems using diffusion models to approximate $p(x \mid y)$ relied on problem-specific architectures (Saharia et al., 2022b; Li et al., 2022; Lugmayr et al., 2022) and required training dedicated generative models for each task (Saharia et al., 2022a; Shi et al., 2022). In contrast, methods that utilize pre-trained diffusion models as priors for posterior sampling offer greater flexibility and are training-free (Kawar et al., 2021; 2022; Chung et al., 2022a;b; Wang et al., 2023), with recent extensions targeting nonlinear inverse problems (Chung et al., 2023; Song et al., 2023a;b; He et al., 2024).

In the context of inverse problems, existing methods can be broadly categorized based on whether they yield a *point estimate* or *multiple estimates*. Existing approaches for posterior sampling focus

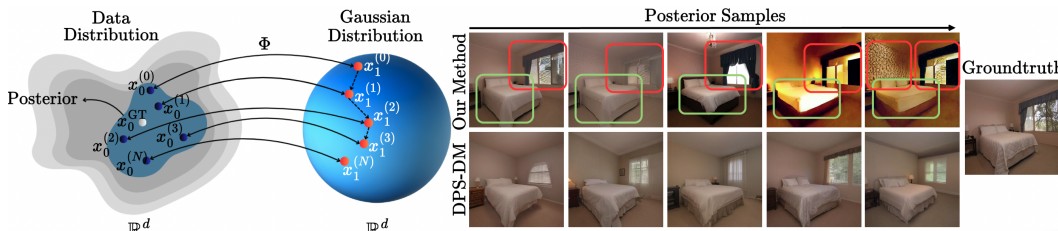

Figure 1: **(Left)**: A schematic representation of posterior sampling via Langevin dynamics in our proposed framework. The sampling process begins with an initial sample $x_1^{(0)}$ from the noise space and maps to data space as $x_0^{(0)}$ using a deterministic mapper $\Phi$ and progressively updates the noise space input to obtain diverse posterior samples. **(Right)**: Posterior samples generated by our method and DPS-DM. Our approach exhibits higher perceptual diversity, capturing variations in high-level features such as lighting, window style, and wall patterns. Uncertain semantic features are highlighted by red boxes, while persistent properties are shown by green boxes.

primarily on providing point estimate solutions (Chung et al., 2023; He et al., 2024; Song et al., 2023a;b), lacking the ability to generate a diverse set of posterior samples efficiently. For instance, a prominent method, Diffusion Posterior Sampling (DPS) (Chung et al., 2023), predominantly produces point estimates for both linear and non-linear inverse problems. DPS leverages the prior $p(x)$ from a diffusion model and employs multiple denoising steps to transform isotropic Gaussian noise into a desired image, guided by the observations $y$. Generating posterior samples using this method requires re-running the entire sampling process using unique instantiations of Gaussian noise, which is computationally prohibitive and inefficient. Therefore, an algorithm that efficiently accumulates samples from the posterior is desirable.

In this work, we propose an efficient framework for posterior sampling by modeling it as an exploration process in the noise space of a pretrained generative model. Specifically, we leverage measurements from the inverse problem to guide the initialization of the noise space, ensuring a more targeted exploration. For sampling, we employ Langevin dynamics directly within the noise space, taking advantage of the one-to-one mapping between noise and data spaces provided by models such as consistency models (Song et al., 2023c). This deterministic mapping eliminates the need for approximating the measurement likelihood, and we establish a theoretical bound on the approximation error for posterior sampling.

Sampling in the noise space allows for a progressive accumulation of posterior samples, enabling efficient exploration and resulting in a diverse set of reconstructions, as demonstrated in Fig-

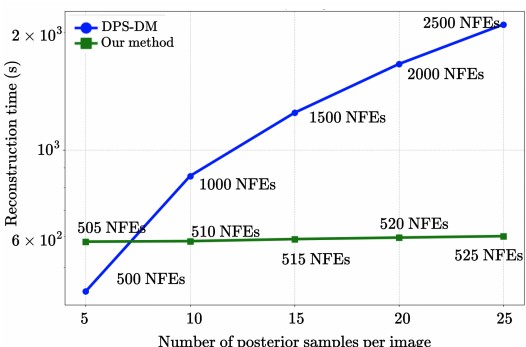

Figure 2: Reconstruction time comparison between DPS-DM and our method for varying numbers of posterior samples. DPS-DM scales poorly with the number of samples, while our method maintains a nearly constant time, demonstrating significantly lower computational cost. The corresponding Number of Function Evaluations (NFEs) (including NFEs for the warmup stage, refer to Section 5) values per image are annotated.

ure 1. Furthermore, Figure 2 illustrates the comparison of reconstruction times between our approach and DPS when generating different numbers of posterior samples per image. While the reconstruction time for DPS increases rapidly with the number of samples, our method incurs only a negligible increase, highlighting its computational efficiency. The key contributions of this work are summarized as follows:

- We present a posterior sampling method defined by Langevin dynamics in the noise space of a pre-trained generative model, enabling efficient accumulation of samples.

- We provide a theoretical guarantee on the posterior sampling approximation error, which is bounded by the approximation error of the prior by the pre-trained generative model.

- Our efficient accumulation of posterior samples facilitates exploration of the posterior, yielding high-fidelity and diverse samples. In experiments, we achieve comparable fidelity to diffusion model posterior sampling methods with superior sample diversity.

**Notation.** We use $\propto$ to stand for the expression of a probability density up to a normalizing constant to enforce integral one, e.g. $p(x) \propto F(x)$ means that $p(x) = F(x)/Z$ where $Z = \int F(x)dx$. For a mapping $T : \mathbb{R}^d \to \mathbb{R}^d$ and a distribution $P$, $T_\# P$ stands for the push-forwarded distribution, that is $T_\# P(A) = P(T^{-1}A)$ for any measurable set $A$. When both $P$ and $T_\# P$ has density, $dP = pdx$, we also use $T_\# p$ to denote the density of $T_\# P$.

## 2 BACKGROUND

**Diffusion models.** Sampling from diffusion models (DMs) is accomplished via simulation of the reverse process corresponding to the forward-time, noising stochastic differential equation (SDE) $dx_t = \mu(x_t, t)dt + \beta(t)dW_t$ (Song et al., 2021b), where $W_t$ is the standard Brownian motion in $\mathbb{R}^d$ and $t \in [0, 1]$. Initialized with data from a data-generating distribution $p_{\text{data}}$, diffusion is typically parameterized such that the terminal distribution of the forward-time SDE is a tractable Gaussian distribution $\gamma$. This SDE shares marginal densities $p_t$ with the *probability flow- (PF-)ODE*:

$$dx_t = \left[ \mu(x_t, t) - \frac{1}{2}\beta(t)^2 \nabla \log p_t(x_t) \right] dt. \tag{1}$$

Score-based generative models are a class of DM which approximate $\nabla \log p_t(x_t)$ with a neural network score model. Given such a model, (1) can be solved in reverse time using numerical ODE integration techniques (Song et al., 2021a; Karras et al., 2022).

**Deterministic diffusion solvers.** In contrast to stochastic DM samplers based on Markov chains (Ho et al., 2020) and SDEs (Song et al., 2021b), deterministic DM solvers primarily focus on simulating the PF-ODE (1). Song et al. (2021a) presented DDIM, an implicit modeling technique yielding a deterministic mapping between noise and data samples. Subsequent works considered alternate, higher-order solvers for the PF-ODE (Karras et al., 2022), yielding high-quality samples in fewer function evaluations.

**Flow models.** Continuous normalizing flows (CNFs) represent another class of ODE-based generative models, using neural networks to approximate the dynamics of a continuous mapping between noise and data (Chen et al., 2018). Recent extensions focus on learning more direct trajectories (Liu et al., 2023b) and simulation-free training (Lipman et al., 2023). As with deterministic diffusion solvers corresponding to the PF-ODE, sampling via these methods requires numerical simulation of an ODE whose dynamics are defined by the neural network model.

**Consistency models.** Efficient ODE simulation is of particular interest for efficient sampling from DMs (Song et al., 2021a; Karras et al., 2022) and CNFs (Lipman et al., 2023). However, fast numerical ODE solvers still require tens of steps to produce high-fidelity samples (Lu et al., 2022; Dockhorn et al., 2022). As a result, score model distillation techniques have arisen to yield fast, effective samplers from the PF-ODE. Consistency models (CMs) are a prominent class of distilled DMs that enable single- and few-step sampling (Song et al., 2023c). CMs learn a mapping $f_\theta$ (parameterized by $\theta$) between a point $x_t$ along the PF-ODE trajectory to the initial state:

$$x_0 = f_\theta(x_t, t) \text{ for } t \in [0, 1], \tag{2}$$

where $x_0$ is a sample from $p_{\text{data}}$. Therefore, single-step sampling can be achieved by sampling $x_1 \sim \gamma$ and evaluating the CM at $x_1$. Multi-step sampling can be achieved by alternating denoising (via evaluation of the CM) and partial noising, trading off efficiency for fidelity.

## 3 METHODOLOGY

Assume that a pre-trained generative model is given, which provides a one-to-one mapping $\Phi$ from the noise space to the data space. The data $x_0$ and noise $x_1$ both belong to $\mathbb{R}^d$, and $x_0 = \Phi(x_1)$. The observation is $y$, and the goal is to sample the data $x_0$ from the posterior distribution $p(x_0|y)$. We derive the posterior sampling of the data vector $x_0$ via that of the noise vector $x_1$, making use of the mapping $\Phi$.

**Likelihood and posterior.**   We consider a general observation model where the conditional law $p(y|x_0)$ is known and differentiable. Define the negative log conditional likelihood as $L_y(x_0) := -\log p(y|x_0)$, which is differentiable with respect to $x_0$ for fixed $y$. A typical case is the inverse problem setting: the *forward* model is

$$y = \mathcal{A}(x_0) + n, \tag{3}$$

where $\mathcal{A} : \mathbb{R}^d \to \mathbb{R}^d$ is the (possibly nonlinear) measurement operator, and $n$ is the additive noise. For fixed $y$, we aim to sample $x_0$ from $p(x_0|y) = p(y|x_0)p(x_0)/p(y) \propto p(y|x_0)p(x_0)$, where $p(x_0)$ is the true prior distribution of all data $x_0$, which we now denote as $p_{\text{data}}$. We also call $p(x_0|y)$ the *true* posterior of $x_0$, donated as

$$p_{0,y}(x_0) := p(x_0|y) \propto p(y|x_0)p_{\text{data}}(x_0). \tag{4}$$

**Posterior approximated via generative model.**   The true data prior $p_{\text{data}}$ is nonlinear and complicated. Let $p_{\text{model}}$ denote the prior distribution approximated by a pre-trained generative model $x_0 = \Phi(x_1)$, where $x_1 \sim \gamma$. A distribution from which samples are easily generated, such as the standard multi-variate Gaussian, is typically chosen for $\gamma$; we choose $\gamma = \mathcal{N}(0, I)$. In other words,

$$p_{\text{data}} \approx p_{\text{model}} = \Phi_{\#}\gamma. \tag{5}$$

Replacing $p_{\text{data}}$ with $p_{\text{model}}$ in (4) gives the *model* posterior of $x_0$, denoted $\tilde{p}_{0,y}$, which approximates the true posterior:

$$p_{0,y}(x_0) \approx \tilde{p}_{0,y}(x_0) \propto p(y|x_0)\Phi_{\#}\gamma(x_0). \tag{6}$$

Because $x_0 = \Phi(x_1)$, we have that $\tilde{p}_{0,y} = \Phi_{\#}\tilde{p}_{1,y}$, where, by a change of variable from (6),

$$\tilde{p}_{1,y}(x_1) \propto p(y|\Phi(x_1))\gamma(x_1). \tag{7}$$

The distribution $\tilde{p}_{1,y}(x_1)$ approximates the posterior distribution $p(x_1|y)$ in the noise space. When $p_{\text{data}} = \Phi_{\#}\gamma$, we have $p_{0,y} = \tilde{p}_{0,y}$ and $p(\cdot|y) = \tilde{p}_{1,y}$. When the generative model prior is inexact, the error in approximating the posterior can be bounded by that in approximating the data prior, see more in Section 4.

**Posterior sampling by Langevin dynamics.**   It is direct to sample the approximated posterior (7) in the noise space using Langevin dynamics. Specifically, since we have $\gamma(x_1) \propto \exp(-\|x_1\|^2/2)$ and $\log p(y|\Phi(x_1)) = -L_y(\Phi(x_1))$, the following SDE of $x_1$ will have $\tilde{p}_{1,y}$ as its equilibrium distribution (proved in Lemma A.1):

$$dx_1 = -(x_1 + \nabla_{x_1} L_y(\Phi(x_1)))dt + \sqrt{2}dW_t. \tag{8}$$

The sampling in the noise space gives the sampling in the data space by the one-to-one mapping of the generative model, namely $x_0 = \Phi(x_1)$.

*Example* 3.1 (Inverse problem with Gaussian noise).  For (3) with white noise, i.e., $n \sim \mathcal{N}(0, \sigma^2 I)$, we have that, with a constant $c$ depending on $(\sigma, d)$,

$$L_y(x_0) = -\log p(y|x_0) = \frac{1}{2\sigma^2}\|y - \mathcal{A}(x_0)\|_2^2 + c.$$

The noise-space SDE (8) can be written as

$$dx_1 = -\left(x_1 + \nabla_{x_1} \frac{\|\mathbf{y} - \mathcal{A}(x_0)\|_2^2}{2\sigma^2}\right) dt + \sqrt{2}dW_t.$$

Given $L_y(x_0)$, standard techniques can be used to sample (overdamped) Langevin dynamics (8). Evaluation of the gradient $\nabla_{x_1} L_y(x_0)$ is the major computational cost, requiring differentiation through the model $\Phi$. One technique to improve sampling efficiency is to employ a warm-start of the SDE integration by letting the minimization-only dynamics (using $\nabla_{x_1} L_y(x_0)$) to converge to a minimum first, especially when the posterior concentrates around a particular point. We postpone the algorithmic details to Section 5.

## 4  THEORY

In this section, we derive the theoretical guarantee of the model posterior $\tilde{p}_{0,y}$ in (6) to the true posterior $p_{0,y}$ in (4), and also extend to the computed posterior $\tilde{p}_{0,y}^S$ by discrete-time SDE integration. The analysis reveals a conditional number which indicates the intrinsic difficulty of the posterior sampling problem. All proofs are in Appendix A.

### 4.1 TOTAL VARIATION (TV) GUARANTEE AND CONDITION NUMBER

Consider the approximation (5), that is, the pre-trained model generates a data prior distribution $\Phi_{\#}\gamma$ that approximates the true data prior $p_{\text{data}}$. We quantify the approximation in TV distance, namely

$$\text{TV}(p_{\text{data}}, \Phi_{\#}\gamma) \leq \varepsilon. \tag{9}$$

Generation guarantee in terms of TV bound has been derived in several flow-based generative model works, such as Chen et al. (2023); Li et al. (2024); Huang et al. (2024) on the PF-ODE of a trained score-based diffusion model (Song et al., 2021b), and Cheng et al. (2024) on the JKO-type flow model (Xu et al., 2023). The following theorem proved in Appendix A shows that the TV distance between the model and true posteriors can be bounded proportional to that between the priors.

**Theorem 4.1** (TV guarantee). *Assuming (9), then* $\text{TV}(p_{0,y}, \tilde{p}_{0,y}) \leq 2\kappa_y \varepsilon$, *where*

$$\kappa_y := \frac{\sup_{x_0} p(y|x_0)}{\int p(y|x)p_{\text{data}}(x)dx}. \tag{10}$$

*Remark* 4.1 ($\kappa_y$ as a condition number). The constant factor $\kappa_y$ is determined by the true data prior $p_{\text{data}}$ and the conditional likelihood $p(y|x_0)$ of the observation, and is independent of the flow model and the posterior sampling method. Thus $\kappa_y$ quantifies an intrinsic "difficulty" of the posterior sampling, which can be viewed as a condition number of the problem.

*Example* 4.1 (Well-conditioned problem). Suppose $p(y|x_0) \leq c_1$ for any $x_0$, and on a domain $\Omega_y$ of the data space,

$$P_{\text{data}}(\Omega_y) \geq \alpha > 0, \quad \text{and} \quad p(y|x_0) \geq c_0 > 0, \; \forall x_0 \in \Omega_y,$$

then we have $\int p(y|x)p_{\text{data}}(x)dx \geq \int_{\Omega_y} p(y|x)p_{\text{data}}(x)dx \geq \alpha c_0$, and then

$$\kappa_y \leq \frac{1}{\alpha}\frac{c_1}{c_0}.$$

This shows that if the observation $y$ can be induced from some cohort of $x_0$ and this cohort is well-sampled by the data prior $p_{\text{data}}$ (the concentration of $p_{\text{data}}$ on this cohort is lower bounded by $\alpha$), plus that the most likely $x_0$ is not too peaked compared to the likelihood of any other $x_0$ within this cohort (the ratio is upper bounded by $c_1/c_0$), then the posterior sampling is well-conditioned.

*Example* 4.2 (Ill-conditioned problem). Suppose $p(y|x_0)$ is peaked at one data value $x_0'$ and almost zero at other places, and this $x_0'$ lies on the tail of the data prior density $p_{\text{data}}$. This means that the integral $\int p(y|x_0)p_{\text{data}}(x_0)dx_0$ has all the contribution on a nearby neighborhood of $x_0'$ on which $p_{\text{data}}$ is small, resulting in a small value on the denominator of (10). Meanwhile, the value of $p(y|x_0')$ is large. In this case, $\kappa_y$ will take a large value, indicating an intrinsic difficulty of the problem. Intuitively, the desired data value $x_0'$ for this observation $y$ is barely represented within the (unconditional) data distribution $p_{\text{data}}$, while the generative model can only learn from $p_{\text{data}}$. Since the pre-trained unconditional generative model does not have enough knowledge of such $x_0'$, it is hard for the conditional generative model (based on the unconditional model) to find such a data value.

### 4.2 TV GUARANTEE OF THE SAMPLED POSTERIOR

Theorem 4.1 captures the approximation error of $\tilde{p}_{0,y}$ to the true posterior, where $\tilde{p}_{0,y}$ is the distribution of data $x_0$ when the noise $x_1$ in noise space achieves the equilibrium $\tilde{p}_{1,y}$ of the SDE (8). In practice, we use a numerical solver to sample the SDE in discrete time. The convergence of discrete-time SDE samplers to its equilibrium distribution has been established under various settings in the literature. Here, we assume that the discrete-time algorithm to sample the Langevin dynamics of $x_1$ outputs $x_1 \sim \tilde{p}_{1,y}^S$, which may differ from but is close to the equilibrium $\tilde{p}_{1,y}$. Specifically, suppose $\text{TV}(\tilde{p}_{1,y}, \tilde{p}_{1,y}^S)$ is bounded by some $\varepsilon_S$.

**Lemma 4.2** (Sampling error). *If* $\text{TV}(\tilde{p}_{1,y}, \tilde{p}_{1,y}^S) \leq \varepsilon_S$, *then* $\text{TV}(\tilde{p}_{0,y}, \tilde{p}_{0,y}^S) \leq \varepsilon_S$.

The lemma is by Data Processing Inequality, and together with Theorem 4.1 it directly leads to the following corollary on the TV guarantee of the sampled posterior.

**Corollary 4.3** (TV of sampled posterior). *Assuming (9) and* $\text{TV}(\tilde{p}_{1,y}, \tilde{p}_{1,y}^S) \leq \varepsilon_S$, *then*

$$\text{TV}(p_{0,y}, \tilde{p}_{0,y}^S) \leq 2\kappa_y \varepsilon + \varepsilon_S.$$

## 5 ALGORITHM

**Numerical integration of the Langevin dynamics.** To numerically integrate the noise-space SDE (8), one can use standard SDE solvers. We adopt the Euler-Maruyama (EM) scheme. Let $\tau > 0$ be the time step, and denote the discrete sequence of $x_1$ as $z^i$, $i = 0, 1, \cdots$. The EM scheme gives, with $\xi^i \sim \mathcal{N}(0, I)$,

$$z^{i+1} = (1 - \tau)z^i - \tau g^i + \sqrt{2\tau}\xi^i, \quad g^i := \nabla_{x_1} L_y(x_0)|_{x_1 = z^i}. \tag{11}$$

See Algorithm 1 for an outline of our approach using EM. However, any general numerical scheme for solving SDEs can be applied; see Table A.4 in Appendix C for a comparison between our method using EM discretization and exponential integrator (EI) (Hochbruck & Ostermann, 2010). An initial value of $z^0$ in the noise space is also required. We adopt a warm-start procedure to initialize sampling; additional details are provided below.

**Computation of $\nabla x_1 L_y(x_0)$.** The computation of the loss gradient depends on the type of generative model representing the mapping $\Phi$. For instance, if $\Phi$ is computed by solving an ODE driven by a normalizing flow, then its gradient can be computed using the adjoint sensitivity method (Chen et al., 2018). If $\Phi$ is a DM or CM sampling scheme, one can backpropagate through the nested function calls to the generative model. Since we use one- or few-step CM sampling to represent $\Phi$ in the experiments, we take the latter approach to compute $\nabla x_1 L_y(x_0)$.

---

**Algorithm 1** Posterior Sampling in Noise Space

**Require:** Forward model $\mathcal{A}$, measurement $y$, loss function $L_y$, pre-trained noise-to-data map $\Phi$, number of steps $N$, step size $\tau$, and initial $x_1^0$
**for** $i = 0, \ldots, N$ **do**
    $x_0^i \leftarrow \Phi(x_1^i)$
    $g^i \leftarrow \nabla_{x_1^i} L_y(x_0^i)$
    $\xi^i \sim \mathcal{N}(0, I)$
    $x_1^{i+1} \leftarrow x_1^i - \tau(x_1^i + g^i) + \sqrt{2\tau}\xi^i$
**end for**
**return** $x_0^1, x_0^2, \ldots, x_0^N$

---

**Choice of initial value and warm-start.** A natural choice for the initial noise space value $z^0$ can be a generic sample $z^0 \sim \gamma$ (the noise space prior). However, while this will correspond to a high-likelihood sample according to the data prior (given a well-trained generative model), it may be far from the data posterior. As such, one may warm-start the sampler by optimizing $L_y(x_0)$ with respect to $x_1$ using standard optimization techniques such as gradient descent or Adam. In all experiments, we warm start sampling using $K$ steps of Adam optimization, initializing EM sampling with $z^0$ being the optimization output. See Appendix B.1 for further detail.

**Computational requirements.** The main computational burden is with respect to the computation of the loss gradient $\nabla_{x_1} L_y(x_0)$, which requires differentiating through the mapping $\Phi$. This can be alleviated by choosing a $\Phi$ which consists of a small number of function evaluations (NFEs). Additional computational burden is due to burn-in/warm start to yield $z^0$, the initial value of EM simulation. Therefore, the total NFEs to simulate $N$ steps of EM (i.e., to yield $N$ samples) is $\eta \cdot (K + N)$, where $\eta$ is the NFEs required to evaluate $\Phi$. However, this burden is amortized over EM sampling, as progressive EM simulation yields increasingly fewer overall NFEs per sample, which asymptotically approaches $\eta$ (the NFEs required to compute $\Phi$). Therefore, we represent $\Phi$ using CM sampling, which can be accomplished for $\eta = 1$ or 2. While multi-step ($\eta > 1$) CM sampling is typically stochastic (Song et al., 2023c), we fix the noise in each step to result in a deterministic mapping. See Appendix B.1 for details.

**Role of EM step size $\tau$.** The step size of EM, $\tau$, controls the time scales over which the Langevin dynamics are simulated with respect to the number of EM steps. Larger $\tau$ results in more rapid exploration of the posterior, potentially leading to more diverse samples over shorter timescales. However, $\tau$ must also be kept small enough to ensure the stability of EM sampling. Therefore, this hyper-parameter provides a degree of control over the diversity of samples provided by the proposed algorithm. Choosing large $\tau$ while maintaining stability can yield diverse samples, potentially revealing particularly uncertain semantic features within the posterior.

## 6 EXPERIMENTS

**Baselines.** We categorize the baselines into two groups. **(1) DM-based methods:** Diffusion Posterior Sampling (DPS) (Chung et al., 2023), Loss-Guided Diffusion (LGD) (Song et al., 2023b), and Manifold-Preserving Guided Diffusion (MPGD) (He et al., 2024). These methods employ stronger priors compared to our approach, making them inherently stronger baselines and rendering the comparison across different backbones unfair.

Table 1: Quantitative comparison of linear image restoration tasks on LSUN-Bedroom (256 x 256) (top table) and ImageNet (64 x 64) (bottom table).

| Method | 8x Super-resolution | | | | Gaussian Deblur | | | | 10% Inpainting | | | |
|---|---|---|---|---|---|---|---|---|---|---|---|---|
| | PSNR ↑ | SSIM ↑ | LPIPS ↓ | FID ↓ | PSNR ↑ | SSIM ↑ | LPIPS↓ | FID ↓ | PSNR ↑ | SSIM ↑ | LPIPS↓ | FID ↓ |
| DPS-DM | 20.4* | 0.538* | 0.470* | 67.7* | 22.1 | 0.589 | 0.407 | 65.3 | 22.4 | 0.634 | 0.417 | 67.7 |
| MPGD-DM | 19.2 | 0.338 | 0.689 | 288 | 23.6* | 0.579 | 0.438 | 85.0 | 15.4 | 0.176 | 0.667 | 221 |
| LGD-DM | 20.1 | 0.529 | 0.483 | 69.3 | 22.2 | 0.590* | 0.371* | 60.1* | 24.7* | 0.742* | 0.289* | 47.3* |
| DPS-CM | 10.7 | 0.077 | 0.758 | 307 | 11.2 | 0.092 | 0.735 | 279 | 19.9 | 0.454 | 0.517 | 128 |
| LGD-CM | 10.5 | 0.072 | 0.764 | 316 | 11.1 | 0.092 | 0.737 | 283 | 19.9 | 0.475 | 0.514 | 134 |
| CMEdit | N/A | | | | N/A | | | | 18.0 | 0.523 | 0.548 | 167 |
| Ours(1-step) | 20.4 | **0.535** | **0.418** | **71.1** | **22.4** | **0.598** | **0.368** | 70.6 | **23.8** | **0.682** | **0.358** | **72.9** |
| Ours(2-step) | **20.5** | 0.534 | 0.433 | 72.2 | 21.3 | 0.554 | 0.421 | **69.2** | 22.2 | 0.611 | 0.419 | 75.6 |

| Method | 4x Super-resolution | | | | Gaussian Deblur | | | | 20% Inpainting | | | |
|---|---|---|---|---|---|---|---|---|---|---|---|---|
| | PSNR ↑ | SSIM ↑ | LPIPS ↓ | FID ↓ | PSNR ↑ | SSIM ↑ | LPIPS↓ | FID ↓ | PSNR ↑ | SSIM ↑ | LPIPS↓ | FID ↓ |
| DPS-DM | 21.0* | 0.531 | 0.310* | 110* | 19.2 | 0.429 | 0.348* | 117* | 22.3* | 0.664* | 0.220* | 89.2* |
| LGD-DM | 21.0* | 0.536* | 0.311 | 114 | 19.6* | 0.432* | 0.352 | 117* | 22.1 | 0.652 | 0.228 | 96.2 |
| DPS-CM | 12.8 | 0.168 | 0.602 | 267 | 9.89 | 0.093 | 0.650 | 334 | 18.9 | 0.470 | 0.371 | 167 |
| LGD-CM | 12.8 | 0.164 | 0.607 | 269 | 10.1 | 0.097 | 0.668 | 363 | 18.7 | 0.451 | 0.380 | 173 |
| Ours(1-step) | 16.9 | **0.418** | **0.388** | **129** | **18.2** | **0.413** | **0.381** | **134** | **20.3** | **0.600** | **0.304** | **124** |
| Ours(2-step) | **18.1** | 0.412 | 0.410 | 151 | 17.2 | 0.347 | 0.435 | 150 | 18.6 | 0.458 | 0.439 | 161 |

**Bold** denotes the best CM method, underline denotes the second best CM method, and * denotes the best DM method.

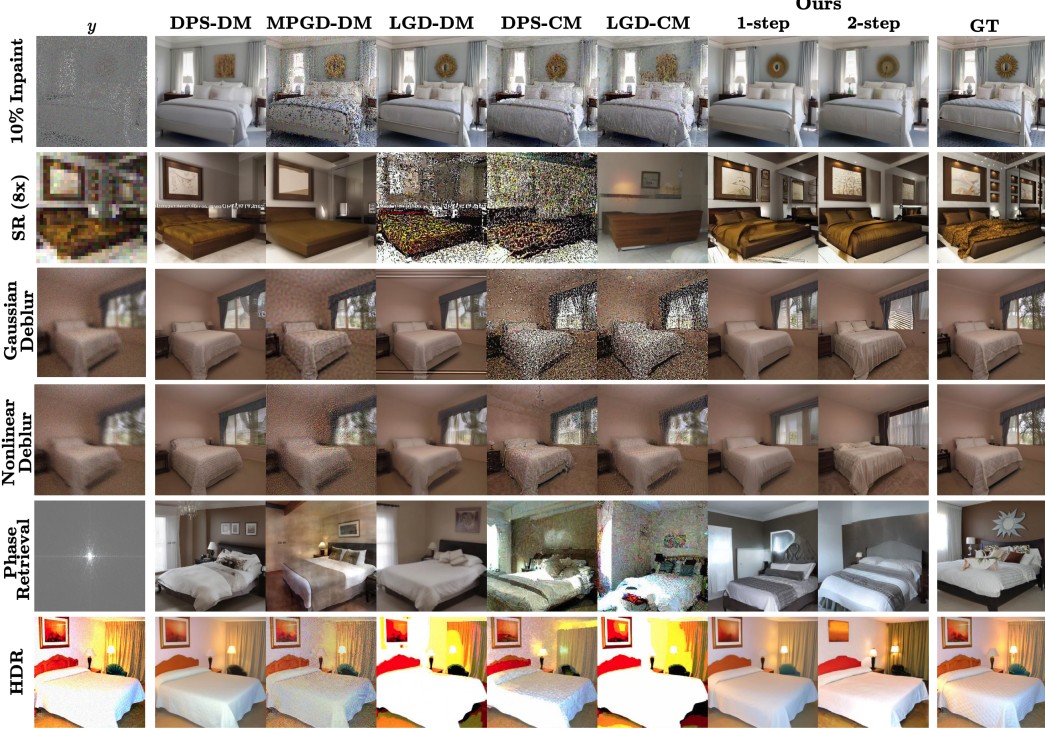

Figure 3: Image reconstructions for the linear and nonlinear tasks on LSUN-Bedroom (256 x 256).

To ensure a fairer comparison, we adopt a second set of baselines: **(2) CM-based methods**, where each DM-based method is adapted to use a consistency model (CM) backbone. Additionally, we include CMEdit, the modified CM sampler from Song et al. (2023c), for linear tasks. All DM baselines use the same EDM model from Song et al. (2023c), and all CM baselines use the corresponding LPIPS-distilled CM. Details and hyper-parameters for each baseline are outlined in Appendix B.2.

**Datasets.** We include experiments on LSUN-Bedroom (256 x 256) (Yu et al., 2024) and ImageNet (64 x 64) (Deng et al., 2009), using 100 validation images for each dataset. All experiments are conducted using the pre-trained CMs from Song et al. (2023c), which were distilled using the LPIPS objective from pre-trained EDM models (Karras et al., 2022). See Appendix B.1 for additional details

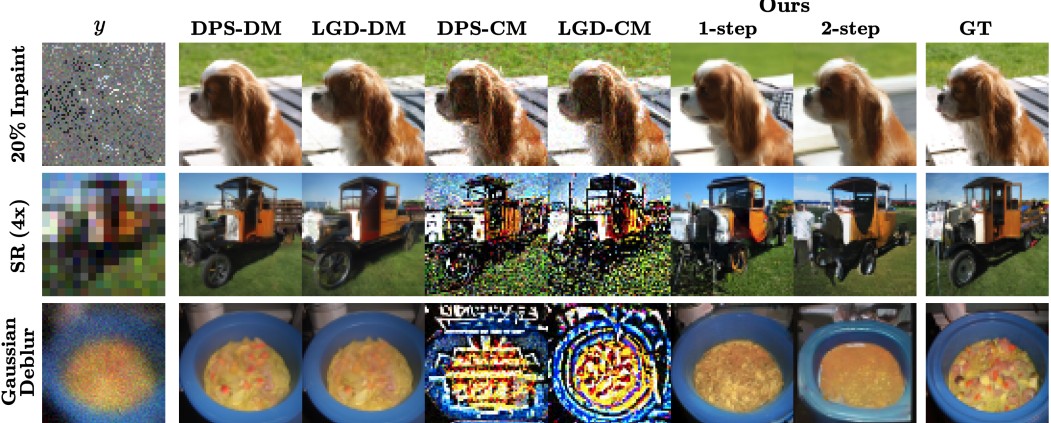

Figure 4: Image reconstructions for the linear tasks on ImageNet (64 x 64).

regarding our method and hyper-parameters. We consider the following linear forward operators for inverse problem tasks: (i) for random mask inpainting, some percentage of the pixels are masked uniformly at random; (ii) for super-resolution, adaptive average pooling is applied; and (iii) for Gaussian deblurring, we use a kernel of $61 \times 61$ pixels with standard deviation 3.0. We also consider nonlinear tasks: (i) nonlinear deblurring using a neural network forward model (Tran et al., 2021); (ii) for phase retrieval, the magnitude of the Fourier coefficients is computed; and (iii) for high dynamic range (HDR) reconstruction, pixel values are multiplied by 2 and again truncated to [-1,1]. All experiments apply Gaussian noise with standard deviation $\sigma = 0.1$ in the measurement space (except for phase retrieval experiments, which use $\sigma = 0.05$). See Appendix B.3 for detailed descriptions of the forward operators. Additional experimental results can be found in Appendices C and D.

**Metrics.** To assess reconstruction fidelity, we compare samples from each method using the Peak Signal-to-Noise Ratio (PSNR), Structural Similarity Index Metric (SSIM), Learned Perceptual Image Patch Similarity (LPIPS), and Fréchet Inception Distance (FID). To assess the diversity of samples, we consider the following metrics: (i) Diversity Score (DS), which is the ratio between the inter- and intra-cluster distances using 6 nearest neighbors clusters of ResNet-50 features, and (ii) Average CLIP Cosine Similarity (CS), which is the average cosine similarity between CLIP embeddings all sample pairs for a given image.

## 6.1 IMAGE RESTORATION RESULTS

**Linear inverse problems.** We quantitatively compare the performance of the proposed approach to the baselines for point-estimate image restoration under linear forward models, where 10 samples are provided by each method for 100 images in the validation datasets. LSUN-Bedroom (256 x 256) results are reported in the top section of Table 1 and our approach is compared to the highest-fidelity baselines on ImageNet (64 x 64) in the bottom section of Table 1. Visual comparisons of point estimates are also visualized in the top three rows of Figure 3 (for LSUN) and in Figure 4 (for ImageNet). Compared with CM baselines, the proposed approach exhibits superior performance in producing high-fidelity candidate solutions to linear inverse problems. This corresponds to improved visual quality, as other CM approaches produce artifacts and poor reconstructions of the ground truth. The proposed method is also competitive against DM baselines, yielding samples of comparable quality both qualitatively and quantitatively.

**Nonlinear inverse problems.** Quantitative comparisons for nonlinear tasks on 100 images from LSUN-Bedroom are displayed in Table 2, where metrics are again computed using 10 samples per image from each method. The proposed method is highly competitive against CM-backbone baselines in all tasks. Moreover, the performance is comparable to that of the DM-backbone baselines. Example reconstructions for each method are visualized in the bottom three rows of Figure 3. Other CM-based methods and MPGD-DM seemingly fail to remove the degradation and noise applied by the forward process, while the proposed method yields samples of visual quality comparable to that of DM baselines. Reconstructions generated using the proposed approach lack the artifacts of CM-backbone baselines while also capturing the fine details present in DM reconstructions. In particular, in the highly degraded and ill-posed phase retrieval task, our method yields samples that are markedly consistent with the ground truth, as PSNR and SSIM values are comparable to those of DM baselines.

Table 2: Quantitative comparison of nonlinear image restoration tasks on LSUN-Bedroom (256 x 256).

| Method | Nonlinear Deblur | | | | Phase Retrieval | | | | HDR Reconstruction | | | |
|---|---|---|---|---|---|---|---|---|---|---|---|---|
| | PSNR ↑ | SSIM ↑ | LPIPS ↓ | FID ↓ | PSNR ↑ | SSIM ↑ | LPIPS ↓ | FID ↓ | PSNR ↑ | SSIM ↑ | LPIPS ↓ | FID ↓ |
| DPS-DM | 21.6 | 0.586 | 0.413 | 75.7* | 10.7 | 0.302 | 0.697* | 90.1 | 21.7* | 0.659* | 0.396* | 69.6* |
| MPGD-DM | 17.0 | 0.194 | 0.683 | 259 | 9.96 | 0.271 | 0.728 | 118 | 20.5 | 0.586 | 0.408 | 73.2 |
| LGD-DM | 22.3* | 0.632* | 0.408* | 106 | 10.8* | 0.351* | 0.709 | 82.0* | 12.4 | 0.459 | 0.560 | 172 |
| DPS-CM | 17.7 | 0.303 | 0.574 | 137 | 10.1 | 0.197 | 0.726 | 195 | 13.5 | 0.405 | 0.597 | 173 |
| MPGD-CM | 13.1 | 0.100 | 0.762 | 306 | 9.39 | 0.111 | 0.786 | 312 | 11.7 | 0.296 | 0.638 | 223 |
| LGD-CM | **21.3** | 0.519 | 0.482 | 163 | 9.36 | 0.113 | 0.767 | 186 | 11.2 | 0.397 | 0.621 | 245 |
| Ours(1-step) | 20.3 | **0.566** | **0.440** | 76.7 | **10.3** | **0.315** | 0.709 | 82.9 | **19.6** | **0.599** | **0.436** | **88.0** |
| Ours(2-step) | 18.7 | 0.501 | 0.492 | **73.3** | 10.2 | 0.309 | **0.708** | **81.4** | 16.6 | 0.481 | 0.532 | 101 |

**Bold** denotes the best CM method, underline denotes the second best CM method, and * denotes the best DM method.

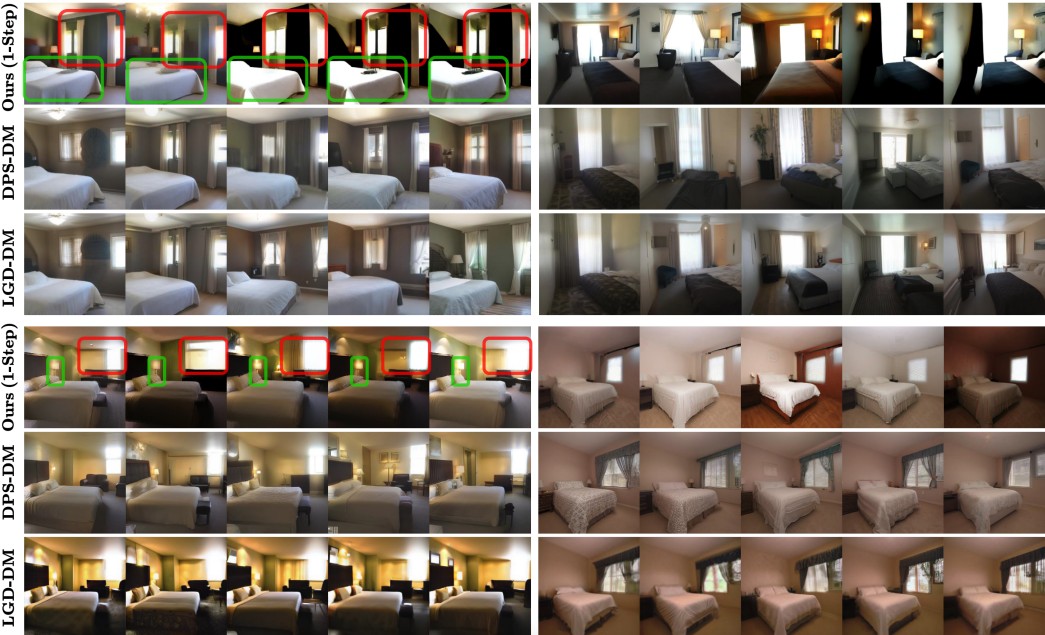

Figure 5: Posterior samples for the inpainitng (10%) (top three rows) and nonlinear deblur (bottom three rows) tasks on LSUN-Bedroom (256 x 256). Green boxes highlight low-uncertainty features and red boxes highlight highly uncertain features.

## 6.2 DIVERSITY OF POSTERIOR SAMPLES

To assess the capacity of the proposed approach to generate diverse samples from the posterior, we conduct additional experiments comparing our method to the strongest baselines: DPS and LGD with a DM backbone. For each of the six (linear and nonlinear) tasks, we generate 25 samples for 100 images from the validation partition of LSUN-Bedroom (256 x 256) via each method. A quantitative comparison of the diversity of the samples from each method is shown in Table 3. Generally, the proposed approach provides competitive to superior performance in diversity metrics compared to DM baselines. Furthermore, visualizing a subset of the posterior samples in the inpainting (top three rows) and nonlinear deblurring (bottom three rows) tasks in Figure 5, one can observe that samples from our method have more clear visual diversity. High-level features of the scene, such as overall lighting or shading, are more variable across our posterior samples. Moreover, our method can identify certain and uncertain semantic features in the candidate reconstructions, as particular features such as windows and lamps have dramatic qualitative variation across the posterior samples from our approach.

Table 3: Quantitative comparison of diversity metrics on linear and non-linear image restoration tasks on LSUN-Bedroom (256 x 256).

| Method | SR(8x) | | Gaussian Deblur | | 10% Inpainting | | Nonlinear Deblur | | Phase Retrieval | | HDR Reconstruction | |
|---|---|---|---|---|---|---|---|---|---|---|---|---|
| | DS ↑ | CS ↓ | DS ↑ | CS ↓ | DS ↑ | CS ↓ | DS ↑ | CS ↓ | DS ↑ | CS ↓ | DS ↑ | CS ↓ |
| DPS-DM | 2.14 | **0.843** | 2.10 | 0.938 | 2.33 | 0.876 | 2.22 | 0.924 | 2.42 | **0.809** | 2.25 | **0.873** |
| LGD-DM | 2.35 | 0.881 | 2.19 | 0.925 | 2.28 | 0.872 | 2.11 | 0.923 | 2.36 | 0.815 | 3.14 | 0.914 |
| Ours(1-step) | **3.01** | 0.879 | **3.26** | 0.997 | **3.15** | 0.869 | **2.80** | 0.912 | **3.08** | 0.914 | 3.09 | 0.927 |
| Ours(2-step) | 2.67 | 0.919 | 2.62 | **0.866** | 2.48 | **0.864** | 2.69 | **0.885** | 2.89 | 0.862 | **3.23** | 0.904 |

**Bold** denotes the best method, underline denotes the second best method.

## 7 RELATED WORKS

**Posterior sampling with generative models.** Diffusion-based inverse problem solvers consist of task-specific frameworks (Saharia et al., 2022b; Li et al., 2022; Lugmayr et al., 2022), optimized approaches (Saharia et al., 2022a; Shi et al., 2022; Liu et al., 2023a), and training-free techniques leveraging pre-trained diffusion priors (Kawar et al., 2021; 2022; Chung et al., 2022a;b; Wang et al., 2023; Chung et al., 2023; Song et al., 2023a;b; He et al., 2024; Dou & Song, 2024). Early training-free methods for solving inverse problems utilize measurement-space projection (Song et al., 2021a; Choi et al., 2021), while others addressed noisy problems via consistency in the spectral domain (Kawar et al., 2021; 2022; Wang et al., 2023) or using manifold constraints (Chung et al., 2022b; He et al., 2024). Recent works consider general noisy and nonlinear inverse problems using an approximation of the measurement likelihood in each generation step (Chung et al., 2023; Song et al., 2023a;b). An emerging area of interest focuses on developing diffusion posterior sampling techniques with provable guarantees (Xu & Chi, 2024; Bruna & Han, 2024). For instance, Xu & Chi (2024) develop an alternating measurement projection/guided diffusion approach for which they provide asymptotic convergence guarantees, while Bruna & Han (2024) utilize tilted transport in linear inverse problems which provably samples the posterior under certain conditions. Diffusion-base posterior sampling works can also be adapted to flow-based models, e.g., Pokle et al. (2023) adapt ΠGDM (Song et al., 2023a) to CNFs. These existing works modify the sampling trajectory of generative priors, requiring repeated simulation of the entire sampling process to produce multiple posterior samples, hindering scalability to many samples. The proposed sampling in the noise space of one- or few-step mappings enables the efficient generation of many posterior samples.

**Guided generation via noise space iteration.** For generative models that provide deterministic mappings between a latent noise space and data, such as GANs (Goodfellow et al., 2014), flows (Chen et al., 2018), and CMs (Song et al., 2023c), optimization of noise can guide generation towards conditional information (Bojanowski et al., 2018; Galatolo. et al., 2021; Patashnik et al., 2021; Asim et al., 2020; Whang et al., 2021; Ben-Hamu et al., 2024). In the GAN literature, this is primarily addressed using text-to-image guided synthesis (Galatolo. et al., 2021; Patashnik et al., 2021) or task-specific objectives (Bojanowski et al., 2018). This type of approach has also been used to solve inverse problems using flow-based models (Asim et al., 2020; Whang et al., 2021); for instance, D-Flow (Ben-Hamu et al., 2024) optimizes with respect to the noise input to CNFs. Our method also iterates in the noise space, simulating Langevin dynamics for posterior sampling instead of optimizing to yield a point estimate. Computing gradients through CNFs is expensive (Chen et al., 2018), requiring at least tens of function evaluation per ODE solution (Lu et al., 2022; Dockhorn et al., 2022). The use of CMs in our approach facilitates computation of the gradient in as few as one call to the neural network, enabling the progressive accumulation of posterior samples during Langevin dynamics simulation.

## 8 DISCUSSION

We have outlined a novel approach for posterior sampling via Langevin dynamics in the noise space of a generative model. Using a CM mapping from noise to data, our posterior sampling provides solutions to general noisy image inverse problems, demonstrating superior reconstruction fidelity to other CM methods and competitiveness with diffusion baselines. A primary limitation of our approach is the low visual quality in some posterior samples. Fidelity drawbacks can be attributed to a relatively poor approximation of the prior by CMs. Future work will focus on improving fidelity of diverse samples, perhaps by using more accurate prior models and adaptive simulation of the SDE. Regardless, our method produces highly diverse samples, representing meaningful semantic uncertainty of data features within the posterior.

## REPRODUCIBILITY STATEMENT

To ensure reproducibility, complete details regarding the implementation of our method are provided in Section 5 and Appendix B.1, including both an algorithmic representation (Algorithm 1) and pseudo-code for a single iteration at the end of Appendix B.1. Hyper-parameters for each experiment are outlined in Tables A.1, A.2, and A.3. Proofs of the theoretical claims made in Sections 3 and 4 can be found in Appendix A.

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

## A   PROOFS

**Lemma A.1.** *The equilibrium distribution of SDE* (8) *is* $\tilde{p}_{1,y}$.

*Proof of Lemma A.1.* Under generic condition, the Langevin dynamics

$$dX_t = -\nabla U(X_t)dt + \sqrt{2}dW_t$$

have the equilibrium $\rho_\infty \propto e^{-U}$. For $\tilde{p}_{1,y}$ in (7) to be the equilibrium, it suffices to verify that

$$\nabla \log \tilde{p}_{1,y} = -(x_1 + \nabla_{x_1} L_y(\Phi(x_1))).$$

This follows by that $\log \gamma(x_1) = -\|x_1\|^2/2 + c$ and $\log p(y|\Phi(x_1)) = -L_y(\Phi(x_1))$. $\qquad\square$

*Proof of Theorem 4.1.* By (4) and (6), we have

$$p_{0,y}(x_0) = \frac{1}{Z_y} p(y|x_0) p_{\text{data}}(x_0), \quad \tilde{p}_{0,y}(x_0) = \frac{1}{\tilde{Z}_y} p(y|x_0) \Phi_{\#}\gamma(x_0),$$

where

$$Z_y := \int p(y|x_0) p_{\text{data}}(x_0) dx_0, \quad \tilde{Z}_y := \int p(y|x_0) \Phi_{\#}\gamma(x_0) dx_0.$$

Then, we have

$$2\,\text{TV}(p_{0,y}, \tilde{p}_{0,y}) = \int |p_{0,y}(x_0) - \tilde{p}_{0,y}(x_0)| dx_0$$

$$\leq \int \frac{1}{Z_y} p(y|x_0) \left| p_{\text{data}}(x_0) - \Phi_{\#}\gamma(x_0) \right| dx_0 + \left| \frac{\tilde{Z}_y - Z_y}{Z_y} \right|. \qquad (A.1)$$

By definition of $\kappa_y$ in (10), we have $\frac{1}{Z_y} p(y|x_0) \leq \kappa_y, \forall x_0$, and thus

$$\int \frac{1}{Z_y} p(y|x_0) \left| p_{\text{data}}(x_0) - \Phi_{\#}\gamma(x_0) \right| dx_0 \leq \kappa_y \int \left| p_{\text{data}}(x_0) - \Phi_{\#}\gamma(x_0) \right| dx_0.$$

Meanwhile, $\tilde{Z}_y - Z_y = \int p(y|x_0)(\Phi_{\#}\gamma(x_0) - p_{\text{data}}(x_0)) dx_0$, and then

$$\frac{|\tilde{Z}_y - Z_y|}{Z_y} \leq \int \frac{1}{Z_y} p(y|x_0) |\Phi_{\#}\gamma(x_0) - p_{\text{data}}(x_0)| dx_0$$

$$\leq \int \kappa_y |\Phi_{\#}\gamma(x_0) - p_{\text{data}}(x_0)| dx_0.$$

Putting back to (A.1), we have

$$2\,\text{TV}(p_{0,y}, \tilde{p}_{0,y}) \leq 2\kappa_y \int |p_{\text{data}}(x_0) - \Phi_{\#}\gamma(x_0)| dx_0 = 4\kappa_y \,\text{TV}(p_{\text{data}}, \Phi_{\#}\gamma),$$

which proves the theorem under (9). □

*Proof of Lemma 4.2.* By that $\tilde{p}_{0,y} = \Phi_{\#}\tilde{p}_{1,y}$, $\tilde{p}_{0,y}^S = \Phi_{\#}\tilde{p}_{1,y}^S$, and Data Processing Inequality. □

*Proof of Corollary 4.3.* By Theorem 4.1, Lemma 4.2, and triangle inequality since TV is half of the $L^1$ norm between two densities. □

# B  EXPERIMENTAL DETAILS

## B.1  DETAILS OF THE PROPOSED APPROACH

**Consistency model generative process.** To represent the map $\Phi$ from noise space to data space, we utilize the pre-trained CMs of Song et al. (2023c) with a 1- or 2-step sampler. For the 2-step sampler, we use standard multistep consistency sampling (Algorithm 1, Song et al. (2023c)), i.e.,

$$x_0 = f_\theta \left( f_\theta(x_T, T) + \sqrt{t^2 - \epsilon^2} z, t \right),$$

where $f_\theta$ is the pre-trained CM, $x_T \leftarrow x_1$, $T = 80$, $\epsilon = 2 \times 10^{-3}$ is a small noise offset, and $t$ is an intermediate "time step" along the PF-ODE trajectory (the "halfway" point). In Song et al. (2023c), $z$ is sampled from the standard Gaussian for each call to $\Phi$. In this work, we sample $z$ once and fix it for all future calls to $\Phi$, which we observe to empirically improve performance.

**Warm-start initialization and sampling.** The posterior sampling process begins with a warm-start initialization consisting of $K$ steps of Adam optimization with learning rate, $\beta_1$, and $\beta_2$ for each experiment outlined in Tables A.1, A.2, and A.3. This is followed by $N$ steps of Langevin dynamics simulation (via EM discretization in the main-text experiments) using step size $\tau$. The NFEs per sample can be computed as $\eta(K + N)/N$, where $\eta$ is the number of steps used for CM generation. All experiments are implemented in PyTorch and are run on a system with NVIDIA A100 GPUs.

See below for a pseudo-code implementation of one iteration of our sampling procedure:

```
1 x1_i = x1_i.requires_grad_()
2 x0_i = denoise(x1_i)
3
4 L = 1 / (2*sigma**2) * torch.norm(y - A(x0_i)) ** 2
5 g_i = torch.autograd.grad(outputs=L, inputs=x1_i)[0]
6
7 x1_i = x1_i - tau * (x1_i + g) + numpy.sqrt(2.*tau) * torch.randn_like(
      x1_i)
8 x1_i = x1_i.detach_()
```

Table A.1: Hyper-parameters for linear and nonlinear image restoration tasks on LSUN-Bedroom (256 x 256).

| Method | 8x Super-resolution | Gaussian Deblur | 10% Inpainting | Nonlinear Deblur | Phase Retrieval | HDR Reconstruction |
|---|---|---|---|---|---|---|
| DPS-DM | $\zeta = 25, N = 100$ | $\zeta = 7, N = 100$ | $\zeta = 25, N = 100$ | $\zeta = 15, N = 100$ | $\zeta = 10, N = 100$ | $\zeta = 5, N = 100$ |
| MPGD-DM | $\zeta = 25, N = 100$ | $\zeta = 15, N = 100$ | $\zeta = 25, N = 100$ | $\zeta = 7, N = 100$ | $\zeta = 7, N = 100$ | $\zeta = 5, N = 100$ |
| LGD-DM | $\zeta = 25, M = 1, N = 100$ | $\zeta = 25, M = 10, N = 100$ | $\zeta = 7, M = 25, N = 100$ | $\zeta = 9, M = 10, N = 100$ | $\zeta = 1, M = 10, N = 100$ | $\zeta = 30, M = 10, N = 100$ |
| DPS-CM | $\zeta = 25, N = 100$ | N/A | $\zeta = 25, N = 100$ | $\zeta = 8, N = 100$ | $\zeta = 9, N = 100$ | $\zeta = 4, N = 100$ |
| MPGD-CM | N/A | N/A | N/A | $\zeta = 15, N = 100$ | $\zeta = 3, N = 100$ | $\zeta = 30, N = 100$ |
| LGD-CM | $\zeta = 25, M = 1, N = 100$ | $\zeta = 7, M = 1, N = 100$ | $\zeta = 5, M = 1, N = 100$ | $\zeta = 15, M = 10, N = 100$ | $\zeta = 0.5, M = 10, N = 100$ | $\zeta = 15, M = 10, N = 100$ |
| Ours(1-step) | Adam: $K = 800, \text{lr} = 5 \times 10^{-3}$ $\beta_1 = 0.9, \beta_2 = 0.999$ EM: $N = 10, \tau = 1 \times 10^{-5}$ | Adam: $K = 800, \text{lr} = 5 \times 10^{-3}$ $\beta_1 = 0.9, \beta_2 = 0.999$ EM: $N = 10, \tau = 1 \times 10^{-6}$ | $K = 800, \text{lr} = 5 \times 10^{-3}$ $\beta_1 = 0.9, \beta_2 = 0.999$ EM: $N = 10, \tau = 1 \times 10^{-5}$ | Adam: $K = 800, \text{lr} = 5 \times 10^{-3}$ $\beta_1 = 0.9, \beta_2 = 0.999$ EM: $N = 10, \tau = 5 \times 10^{-6}$ | Adam: $K = 200, \text{lr} = 1 \times 10^{-3}$ $\beta_1 = 0.9, \beta_2 = 0.999$ EM: $N = 10, \tau = 1 \times 10^{-6}$ | $K = 800, \text{lr} = 5 \times 10^{-3}$ $\beta_1 = 0.9, \beta_2 = 0.999$ EM: $N = 10, \tau = 1 \times 10^{-6}$ |
| Ours(2-step) | Adam: $K = 800, \text{lr} = 5 \times 10^{-3}$ $\beta_1 = 0.9, \beta_2 = 0.999$ EM: $N = 10, \tau = 1 \times 10^{-5}$ | Adam: $K = 800, \text{lr} = 5 \times 10^{-3}$ $\beta_1 = 0.9, \beta_2 = 0.999$ EM: $N = 10, \tau = 1 \times 10^{-7}$ | Adam: $K = 800, \text{lr} = 5 \times 10^{-3}$ $\beta_1 = 0.9, \beta_2 = 0.999$ EM: $N = 10, \tau = 1 \times 10^{-5}$ | Adam: $K = 500, \text{lr} = 5 \times 10^{-3}$ $\beta_1 = 0.9, \beta_2 = 0.999$ EM: $N = 10, \tau = 5 \times 10^{-6}$ | Adam: $K = 200, \text{lr} = 1 \times 10^{-3}$ $\beta_1 = 0.9, \beta_2 = 0.999$ EM: $N = 10, \tau = 1 \times 10^{-6}$ | Adam: $K = 500, \text{lr} = 5 \times 10^{-3}$ $\beta_1 = 0.9, \beta_2 = 0.999$ EM: $N = 10, \tau = 1 \times 10^{-6}$ |

Table A.2: Hyper-parameters for linear image restoration tasks on ImageNet (64 x 64).

| Method | 4x Super-resolution | Gaussian Deblur | 20% Inpainting |
|---|---|---|---|
| DPS-DM | $\zeta = 20, N = 100$ | $\zeta = 15, N = 100$ | $\zeta = 30, N = 100$ |
| LGD-DM | $\zeta = 3, M = 10, N = 100$ | $\zeta = 1, M = 10, N = 100$ | $\zeta = 5, M = 10, N = 100$ |
| DPS-CM | $\zeta = 30, N = 100$ | $\zeta = 30, N = 100$ | $\zeta = 25, N = 100$ |
| LGD-CM | $\zeta = 3, M = 10, N = 100$ | $\zeta = 7, M = 10, N = 100$ | $\zeta = 6, M = 10, N = 100$ |
| Ours(1-step) | Adam: $K = 800, \text{lr} = 1 \times 10^{-2}$ $\beta_1 = 0.9, \beta_2 = 0.999$ EM: $N = 10, \tau = 5 \times 10^{-4}$ | Adam: $K = 800, \text{lr} = 1 \times 10^{-2}$ $\beta_1 = 0.9, \beta_2 = 0.999$ EM: $N = 10, \tau = 3 \times 10^{-5}$ | $K = 800, \text{lr} = 1 \times 10^{-2}$ $\beta_1 = 0.9, \beta_2 = 0.999$ EM: $N = 10, \tau = 1 \times 10^{-4}$ |
| Ours(2-step) | Adam: $K = 500, \text{lr} = 5 \times 10^{-2}$ $\beta_1 = 0.9, \beta_2 = 0.999$ EM: $N = 10, \tau = 1 \times 10^{-4}$ | Adam: $K = 500, \text{lr} = 5 \times 10^{-2}$ $\beta_1 = 0.9, \beta_2 = 0.999$ EM: $N = 10, \tau = 3 \times 10^{-5}$ | Adam: $K = 500, \text{lr} = 5 \times 10^{-2}$ $\beta_1 = 0.9, \beta_2 = 0.999$ EM: $N = 10, \tau = 1 \times 10^{-4}$ |

Table A.3: Hyper-parameters for linear and nonlinear diversity experiments on LSUN-Bedroom (256 x 256).

| Method | 8x Super-resolution | Gaussian Deblur | 10% Inpainting | Nonlinear Deblur | Phase Retrieval | HDR Reconstruction |
|---|---|---|---|---|---|---|
| DPS-DM | $\zeta = 7, N = 100$ | $\zeta = 7, N = 100$ | $\zeta = 7, N = 100$ | $\zeta = 5, N = 100$ | $\zeta = 5, N = 100$ | $\zeta = 1, N = 100$ |
| LGD-DM | $\zeta = 15, M = 1, N = 100$ | $\zeta = 5, M = 1, N = 100$ | $\zeta = 15, M = 1, N = 100$ | $\zeta = 4, M = 10, N = 100$ | $\zeta = 0.5, M = 10, N = 100$ | $\zeta = 10, M = 10, N = 100$ |
| Ours(1-step) | Adam: $K = 400, \text{lr} = 5 \times 10^{-3}$ $\beta_1 = 0.9, \beta_2 = 0.999$ EM: $N = 10, \tau = 4 \times 10^{-4}$ | Adam: $K = 600, \text{lr} = 5 \times 10^{-3}$ $\beta_1 = 0.9, \beta_2 = 0.999$ EM: $N = 10, \tau = 1 \times 10^{-6}$ | $K = 600, \text{lr} = 5 \times 10^{-3}$ $\beta_1 = 0.9, \beta_2 = 0.999$ EM: $N = 10, \tau = 1 \times 10^{-4}$ | Adam: $K = 800, \text{lr} = 5 \times 10^{-3}$ $\beta_1 = 0.9, \beta_2 = 0.999$ EM: $N = 25, \tau = 7.5 \times 10^{-6}$ | Adam: $K = 200, \text{lr} = 1 \times 10^{-3}$ $\beta_1 = 0.9, \beta_2 = 0.999$ EM: $N = 25, \tau = 3 \times 10^{-6}$ | $K = 800, \text{lr} = 5 \times 10^{-3}$ $\beta_1 = 0.9, \beta_2 = 0.999$ EM: $N = 25, \tau = 3 \times 10^{-6}$ |
| Ours(2-step) | Adam: $K = 600, \text{lr} = 5 \times 10^{-3}$ $\beta_1 = 0.9, \beta_2 = 0.999$ EM: $N = 10, \tau = 4 \times 10^{-4}$ | Adam: $K = 600, \text{lr} = 5 \times 10^{-3}$ $\beta_1 = 0.9, \beta_2 = 0.999$ EM: $N = 10, \tau = 1 \times 10^{-5}$ | Adam: $K = 800, \text{lr} = 5 \times 10^{-3}$ $\beta_1 = 0.9, \beta_2 = 0.999$ EM: $N = 10, \tau = 1 \times 10^{-4}$ | Adam: $K = 500, \text{lr} = 5 \times 10^{-3}$ $\beta_1 = 0.9, \beta_2 = 0.999$ EM: $N = 25, \tau = 3 \times 10^{-6}$ | Adam: $K = 500, \text{lr} = 1 \times 10^{-3}$ $\beta_1 = 0.9, \beta_2 = 0.999$ EM: $N = 25, \tau = 3 \times 10^{-6}$ | Adam: $K = 500, \text{lr} = 5 \times 10^{-3}$ $\beta_1 = 0.9, \beta_2 = 0.999$ EM: $N = 25, \tau = 3 \times 10^{-6}$ |

## B.2 DETAILS OF THE BASELINES

The baseline methods conduct $t = 1, \ldots, N$ Euler steps for sampling. All methods require a denoiser to provide $x_0 \approx \hat{x}_0(x_t)$ at each sampling step $t$, which is achieved using either a pre-trained EDM (Karras et al., 2022) or CM (Song et al., 2023c), both obtained from Song et al. (2023c) for each dataset.

**Diffusion Posterior Sampling (DPS).** DPS (Chung et al., 2023) utilizes the denoiser corresponding to a pre-trained DM to approximate the measurement likelihood gradient at each step of DM sampling. At each state $x_t$ along the diffusion sampling trajectory, a score-base diffusion model can provide a predicted $\hat{x}_0(x_t)$, which can be used to compute $\nabla_{x_t} p(y|\hat{x}_0)$ via differentiation through the score-based model. In DPS, each step of diffusion sampling is adjusted by this gradient with weight $\zeta$, i.e., $x_{t-1} \leftarrow x_{t-1} - \zeta \nabla_{x_t} p(y|\hat{x}_0)$.

**Manifold Preserving Guided Diffusion (MPGD).** MPGD (He et al., 2023) computes the gradient of the measurement likelihood in the denoised space rather than with respect to $x_t$ at each step, taking a gradient step in $\hat{x}_0$ before updating the diffusion iterate. That is, MPGD conducts the update $\hat{x}_0 \leftarrow \hat{x}_0(x_t) - \zeta \nabla_{\hat{x}_0} p(y|\hat{x}_0(x_t))$, which can then be use to yield $x_{t-1}$ at each step. MPGD also provides an optional manifold projection step which utilizes pre-trained autoencoders to ensure $\hat{x}_0$ remains on the data manifold. For a fair comparison, we only consider MPGD without manifold projection in this work.

**Loss Guided Diffusion (LGD).** LGD (Song et al., 2023b) aims to improve the approximation of $p(y|x_0)$ at each step along the sampling trajectory via a Monte Carlo approach. Viewing $p(y|\hat{x}_0)$ in DPS as a delta distribution approximation of $p(y|x_0)$ about $\hat{x}_0$, LGD instead computes the log-mean-exponential of $p(y|\hat{x}_0^{(m)})$ for $m = 1, \ldots, M$ perturbed copies of $\hat{x}_0$. That is, $p(\hat{x}_0|x_t) \sim \mathcal{N}(\hat{x}_0(x_t), r_t^2 I)$, where $r_t = \beta_t/\sqrt{1 + \beta_t^2}$. The weighted (by $\zeta$) Monte Carlo gradient $\nabla_{x_t} \log \left( \frac{1}{M} \sum_{m=1}^{M} \exp \left( p \left( y|\hat{x}_0^{(m)} \right) \right) \right)$ is then used to adjust $x_{t-1}$, as in DPS.

### B.3 DEGRADATIONS AND FORWARD OPERATORS

In all experiments, pixel values are scaled from [-1, 1] (as in Song et al. (2023c)) before application of forward operators. The details of the measurement likelihoods corresponding to each forward operator are outlined below. All methods use $\sigma = 0.1$, except for phase retrieval, which uses $\sigma = 0.05$.

**Super-resolution.** The super-resolution task is defined by the following measurement likelihood:

$$y \sim \mathcal{N}(y|\text{AvgPool}_f(x), \sigma^2 I),$$

where $\text{AvgPool}$ represents 2D average pooling by a factor $f$.

**Gaussian deblur.** Gaussian blur is defined by a block Hankel matrix $C^\psi$ representing convolution of $x$ with kernel $\psi$:

$$y \sim \mathcal{N}(y|C^\psi x, \sigma^2 I).$$

We consider a 61 x 61 Gaussian kernel with standard deviation of 3.0, as in Chung et al. (2023).

**Inpainting.** The measurement likelihood corresponding to $p\%$ inpainting is a function of a mask $P$ with (1-$p$)% uniformly random 0 values:

$$y \sim \mathcal{N}(y|Px, \sigma^2 I).$$

**Nonlinear deblur.** Following Chung et al. (2023), the forward nonlinear blur operator is a pre-trained neural network $\mathcal{F}_\phi$ to approximate the integration of non-blurry images over a short time frame given a single sharp image (Tran et al., 2021). Therefore, the measurement likelihood is as follows:

$$y \sim \mathcal{N}(y|\mathcal{F}_\phi(x), \sigma^2 I).$$

**Phase retrieval.** The forward operator of the phase retrieval task takes the absolute value of the 2D Discrete Fourier Transform $F$ applied to $x$: $|Fx|$. However, since this task is known to be highly ill-posed (Hayes, 1982; Chung et al., 2023), an oversampling matrix $P$ is also applied (with oversampling ratio 1 in this work):

$$y \sim \mathcal{N}(y||FPx|, \sigma^2 I).$$

**High dynamic range reconstruction.** In the HDR forward model, pixel values are scaled by a factor of 2 before truncation back to the range [-1, 1]. Therefore, the measurement likelihood is as follows:

$$y \sim \mathcal{N}(y|\text{clip}(2x, -1, 1), \sigma^2 I),$$

where $\text{clip}(\cdot, -1, 1)$ truncates all input values to the range [-1, 1].

## C  ADDITIONAL EXPERIMENTS

**Numerical SDE solver comparison.**    Alternative numerical methods to EM (11) can be applied to discretize the Langevin dynamics SDE, such as the exponential integrator (EI) (Hochbruck & Ostermann, 2010). The EI scheme discretizes the nonlinear drift term $g^i = \nabla_{x_1} L_y(x_0)|_{x_1=z^i}$ and integrates the continuous-time dynamics arising from the linear term:

$$z^{i+1} = e^{-\tau}z^i - (1 - e^{-\tau})g^i + \sqrt{1 - e^{-2\tau}}\xi^i,$$

where $\xi^i \sim \mathcal{N}(0, I)$. In Table A.4, quantitative comparison between our method using EM versus EI is shown on generating 10 samples for 100 images from the LSUN-Bedroom validation dataset, where the forward operator is nonlinear blurring. The same hyper-parameters are used for both methods, which are outlined in Table A.1. In this case, there is a marginal improvement in most metrics when using the EI scheme.

Table A.4: Comparison between our method with EM and EI integration on the nonlinear deblur task on LSUN-Bedroom (256 x 256).

| Method | PSNR ↑ | SSIM ↑ | LPIPS ↓ | FID ↓ |
|---|---|---|---|---|
| Ours-EM(1-step) | 20.3 | 0.566 | 0.440 | 76.7 |
| Ours-EM(2-step) | 18.7 | 0.501 | 0.492 | 73.3 |
| Ours-EI(1-step) | 20.5 | 0.569 | 0.437 | 76.3 |
| Ours-EI(2-step) | 18.7 | 0.504 | 0.491 | 74.2 |

## D  ADDITIONAL QUALITATIVE RESULTS

Visualizations of additional reconstructions from our method corresponding to the linear and nonlinear experiments from Section 6.1 can be found in Figures A.1, A.2, A.3, A.4, and A.5. Additionally, diverse sets of samples from our one-step / two-step CM method corresponding to the experiments of Section 6.2 are visualized in Figures A.6, A.7, A.8, A.9, A.10, and A.11. Finally, diverse samples via the linear tasks on ImageNet (64 x 64) are shown in Figures A.12, A.13, and A.14. In these experiments, we use the one-step CM sampler with the same hyper-parameters as in Table A.2, but with $\tau = 4 \times 10^{-4}$ for inpainting, $\tau = 9 \times 10^{-4}$ for super-resolution, and $\tau = 5 \times 10^{-5}$ for Gaussian deblur.

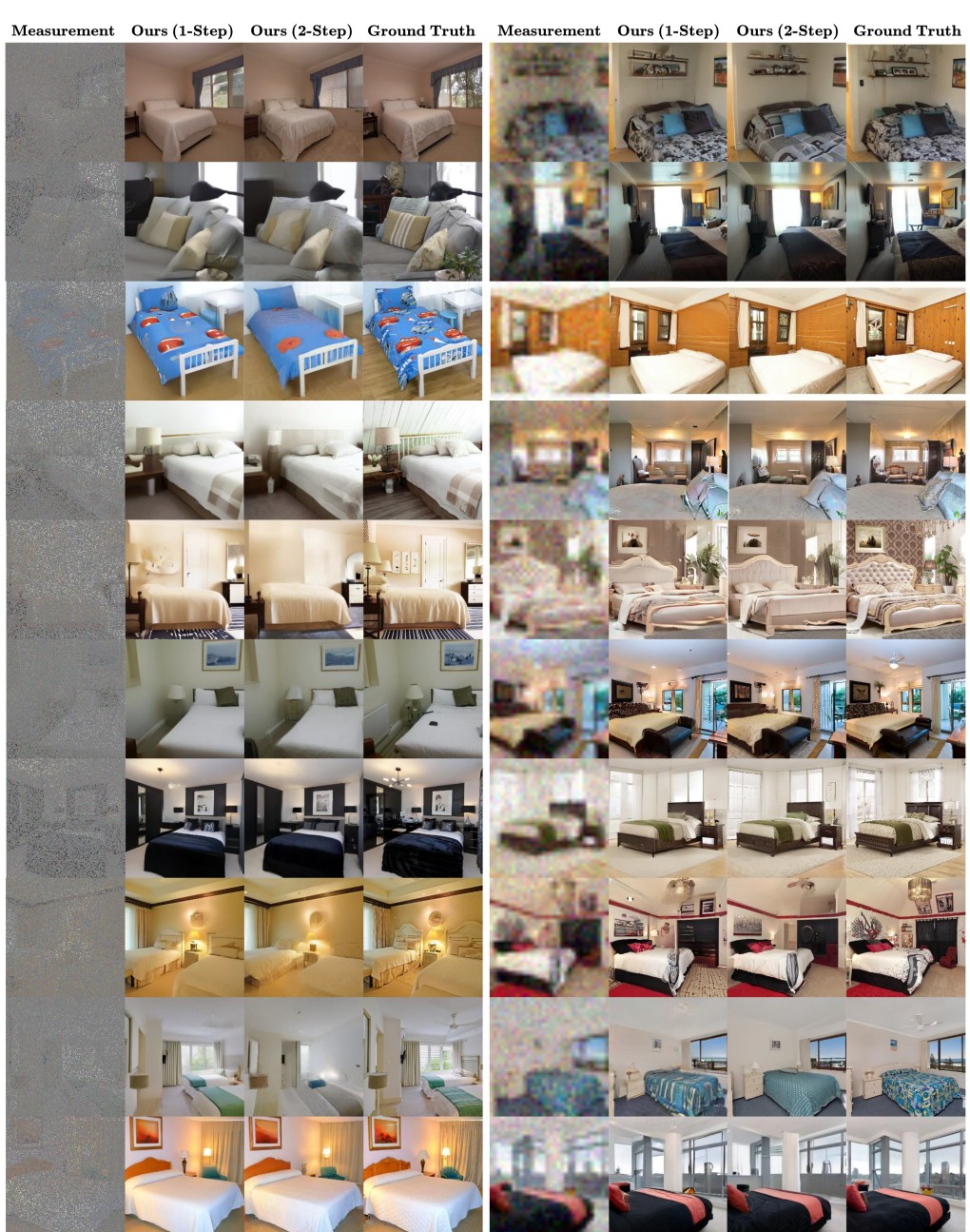

Figure A.1: Additional image reconstructions for inpainting (left) and 8x super-resolution (right) on LSUN-Bedroom (256 x 256).

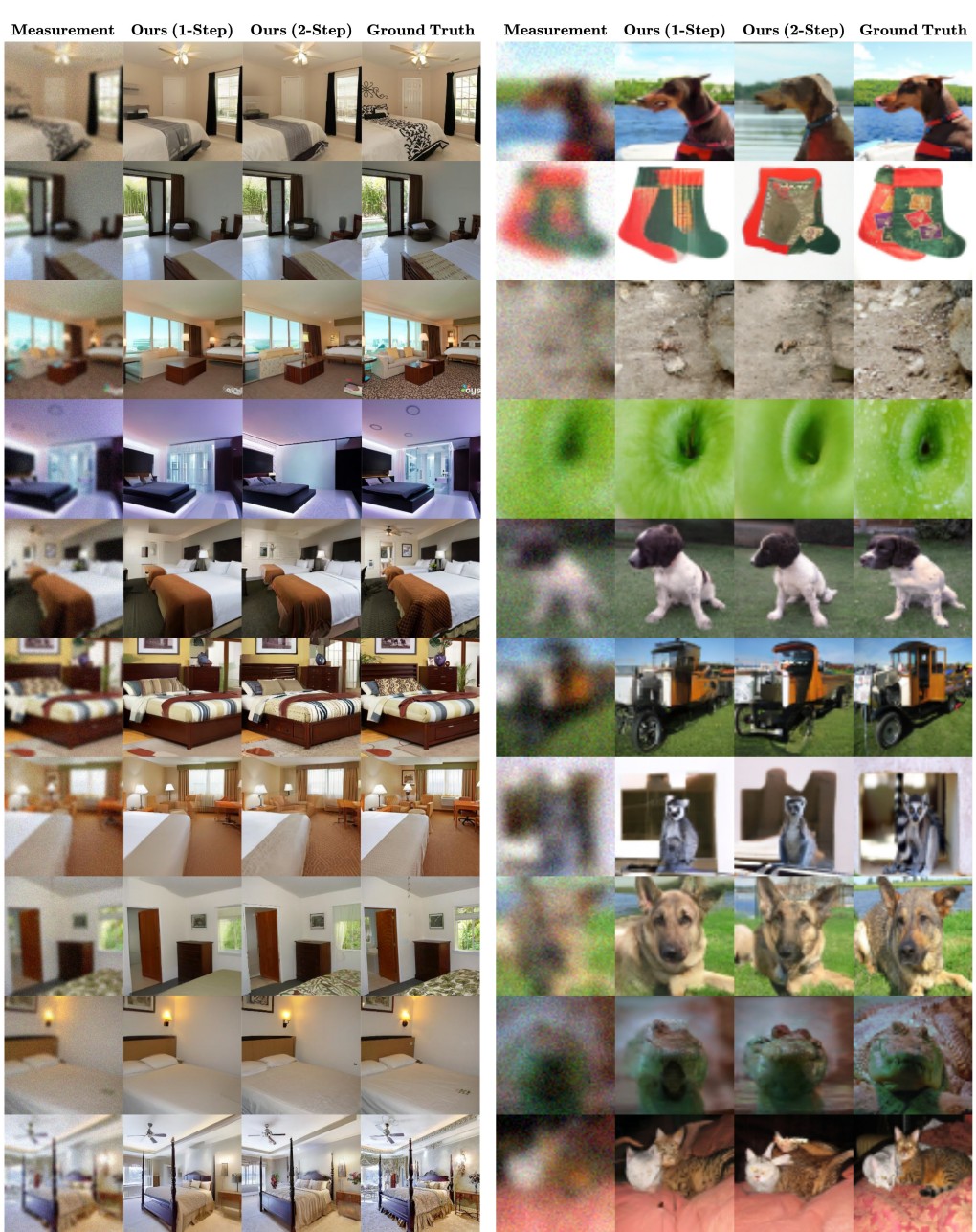

Figure A.2: Additional image reconstructions for Gaussian Deblurring on LSUN-Bedroom (256 x 256) (left) and ImageNet (64 x 64) (right).

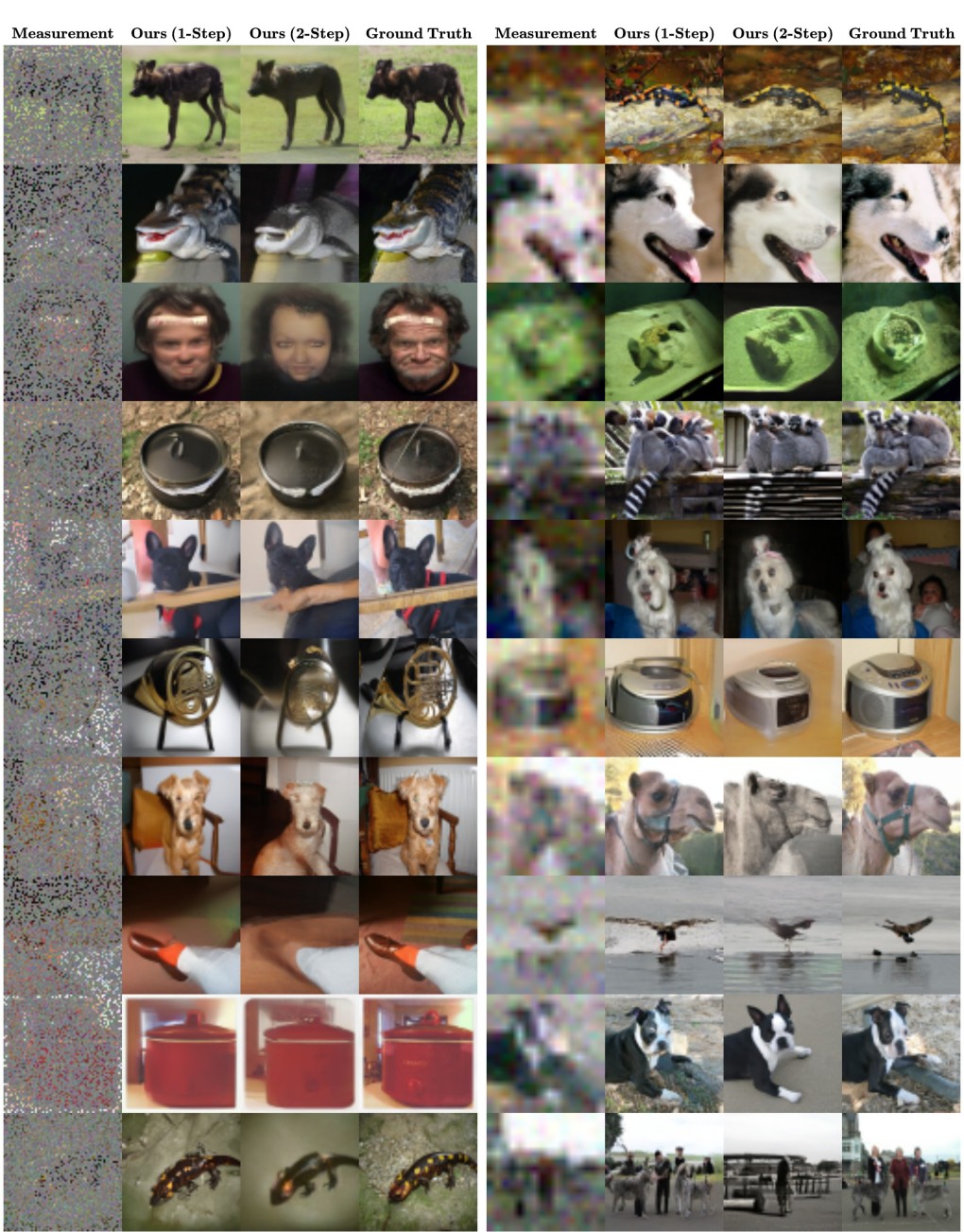

Figure A.3: Additional image reconstructions for inpainting (left) and 4x super-resolution (right) on ImageNet (64 x 64).

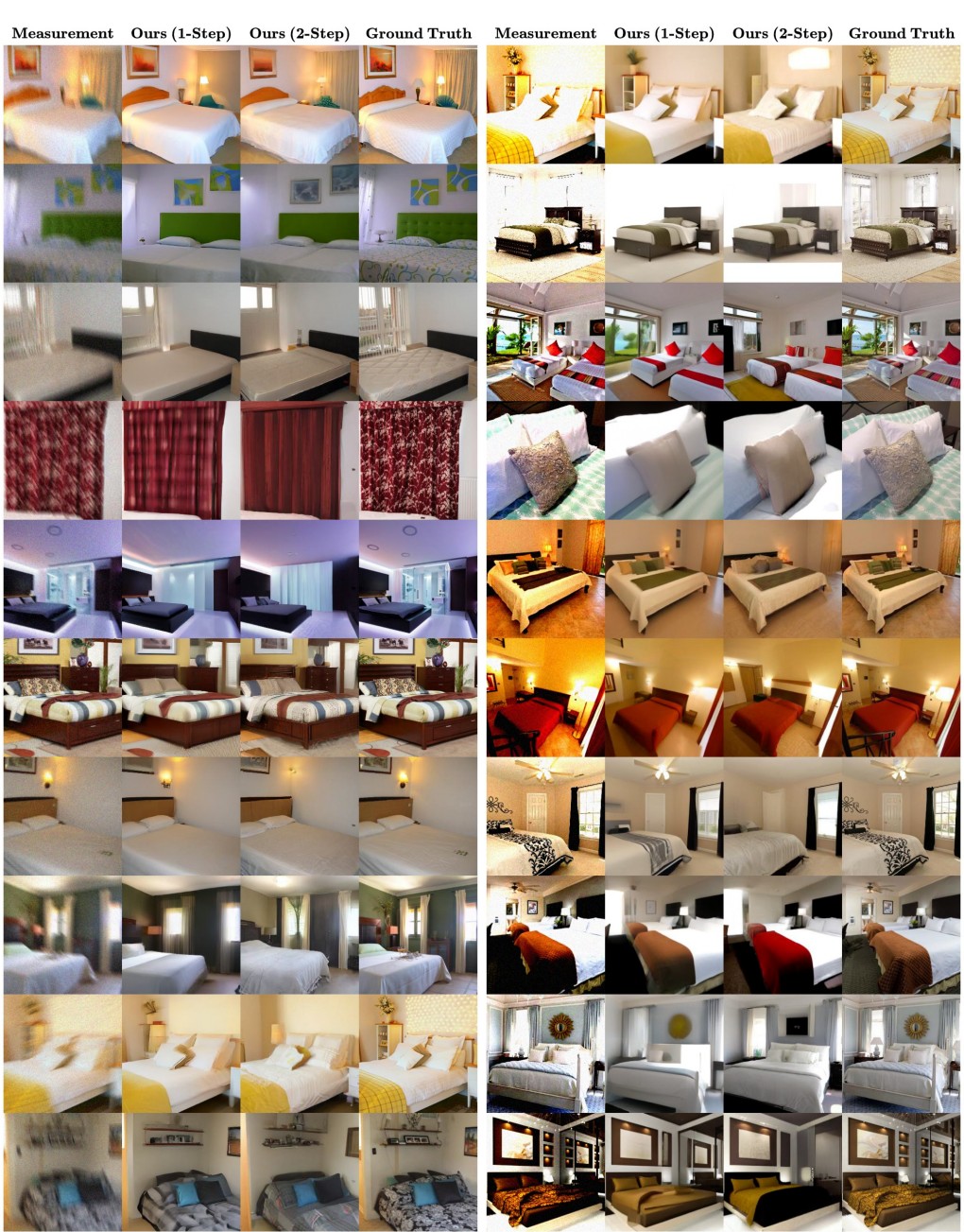

Figure A.4: Additional image reconstructions for nonlinear deblur (left) and HDR reconstruction (right) on LSUN-Bedroom (256 x 256).

Figure A.5: Additional image reconstructions for phase retrieval on LSUN-Bedroom (256 x 256).

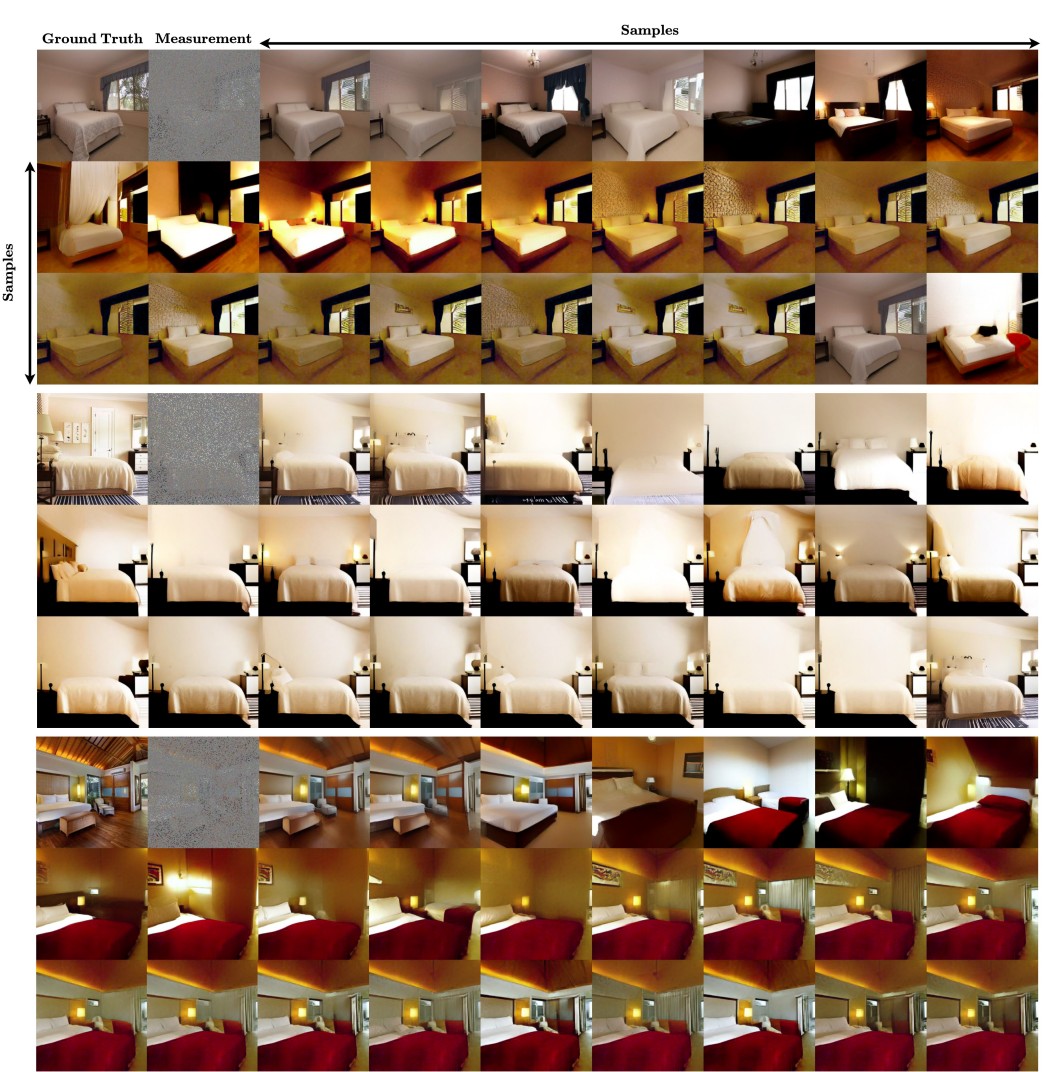

Figure A.6: Additional sets of samples for Inpainting (10%) on LSUN-Bedroom (256 x 256).

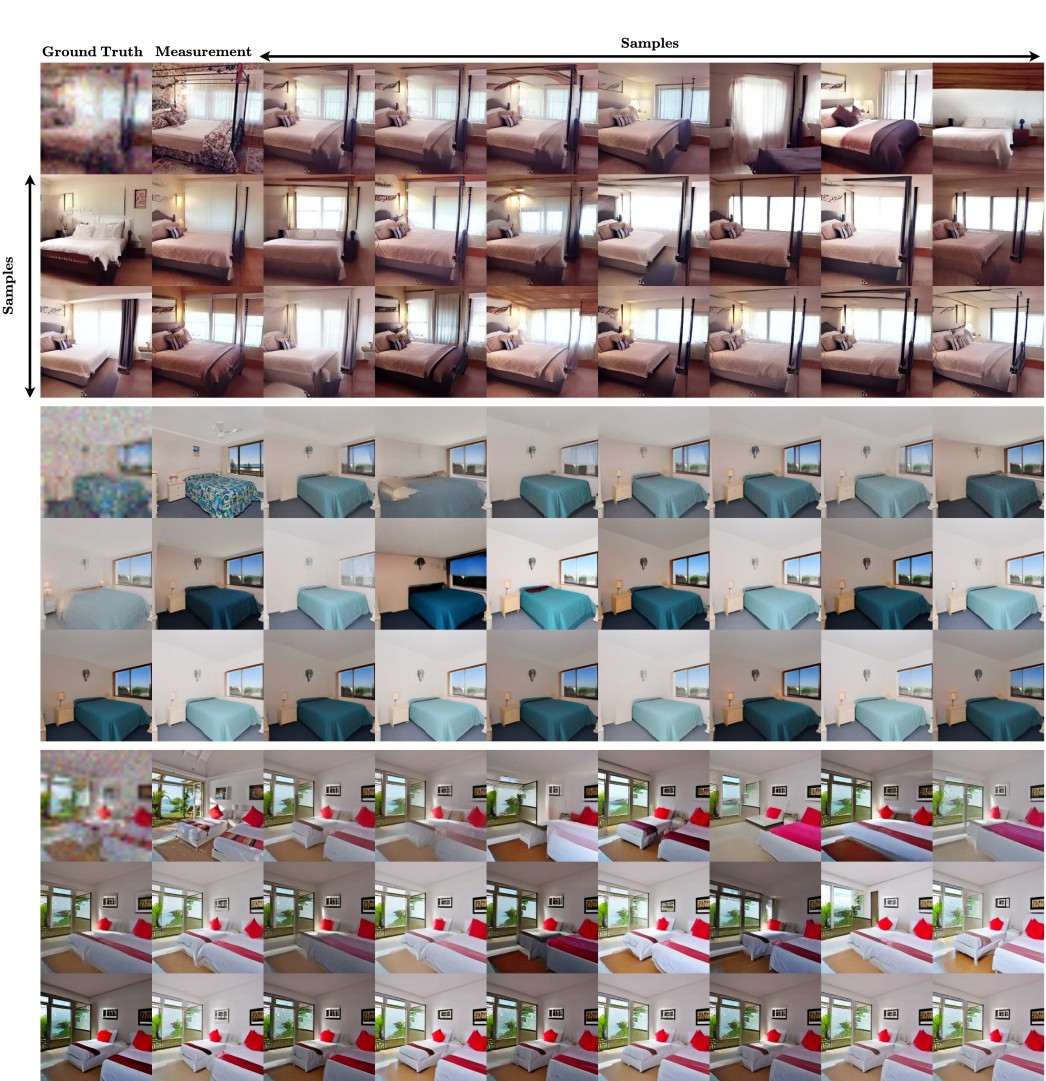

Figure A.7: Additional sets of samples for SR (8x) on LSUN-Bedroom (256 x 256).

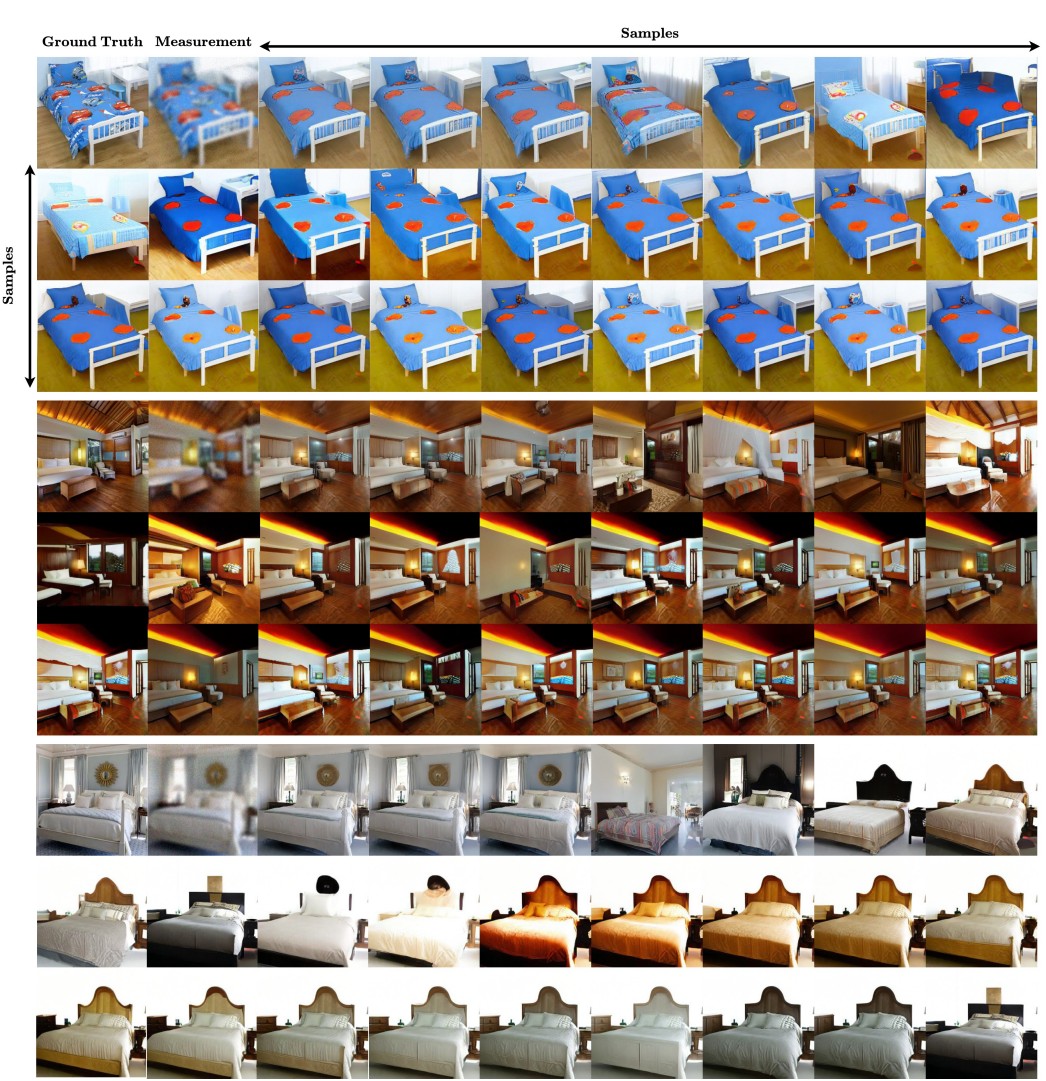

Figure A.8: Additional sets of samples for SR (8x) on LSUN-Bedroom (256 x 256) for 2-step method.

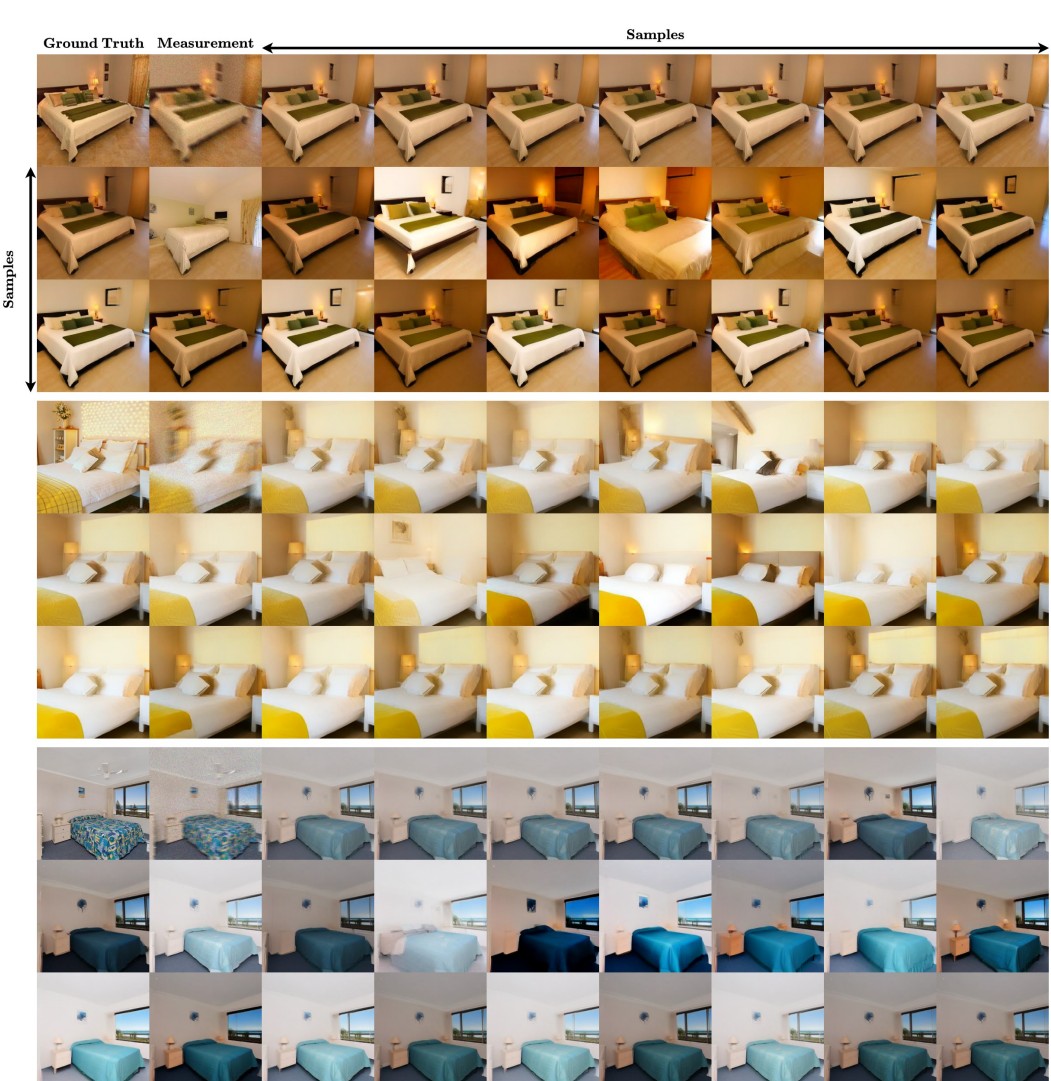

Figure A.9: Additional sets of samples for nonlinear deblur on LSUN-Bedroom (256 x 256).

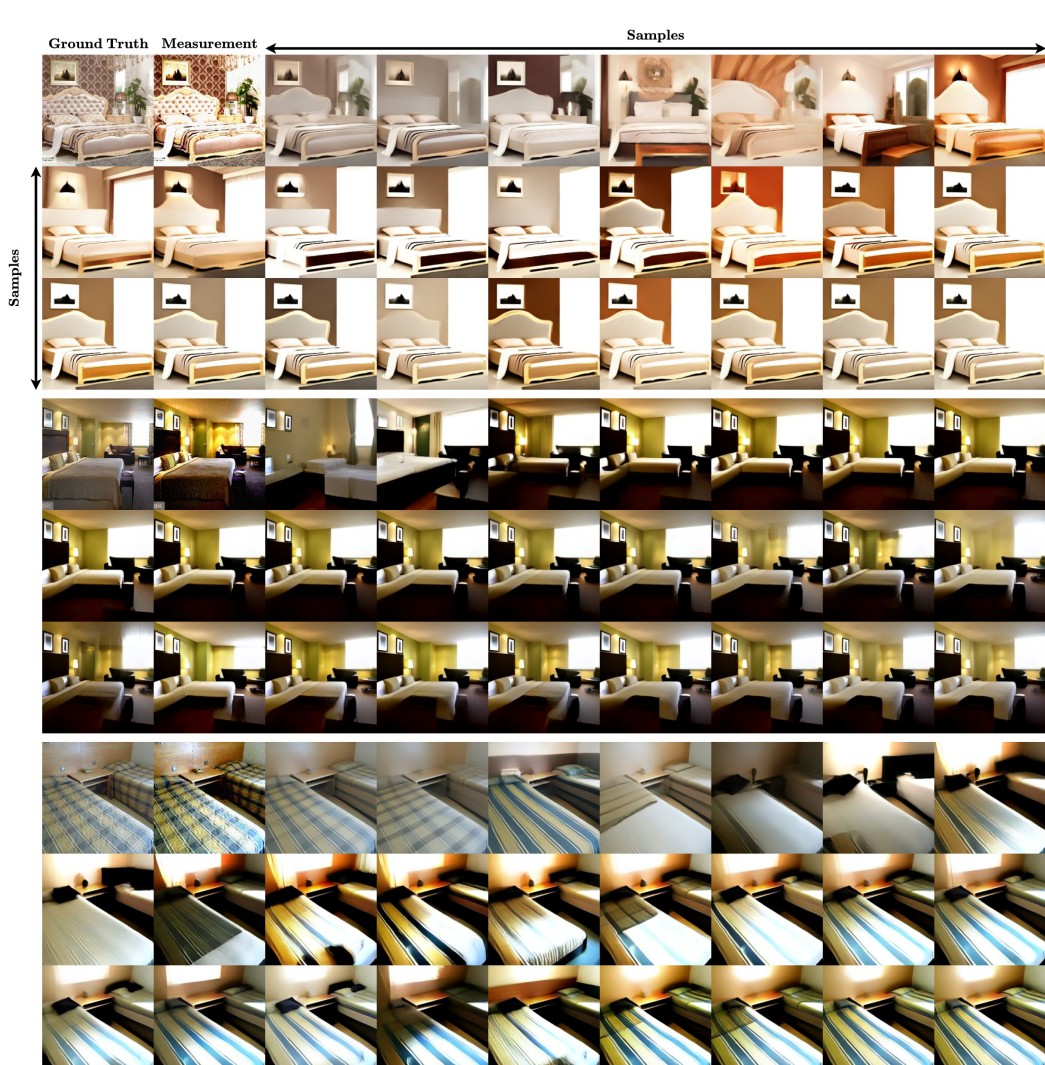

Figure A.10: Additional sets of samples for HDR reconstruction on LSUN-Bedroom (256 x 256).

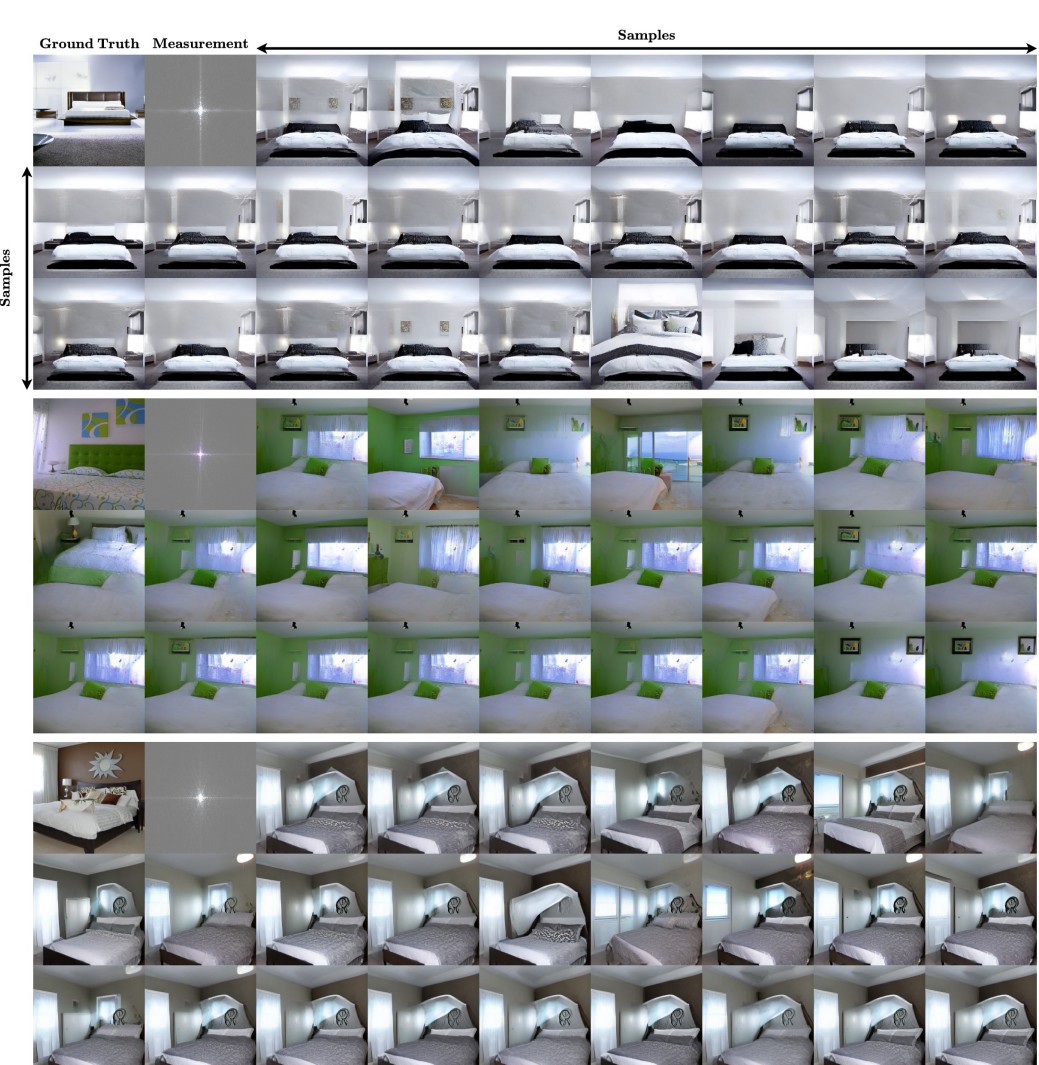

Figure A.11: Additional sets of samples for phase retrieval on LSUN-Bedroom (256 x 256).

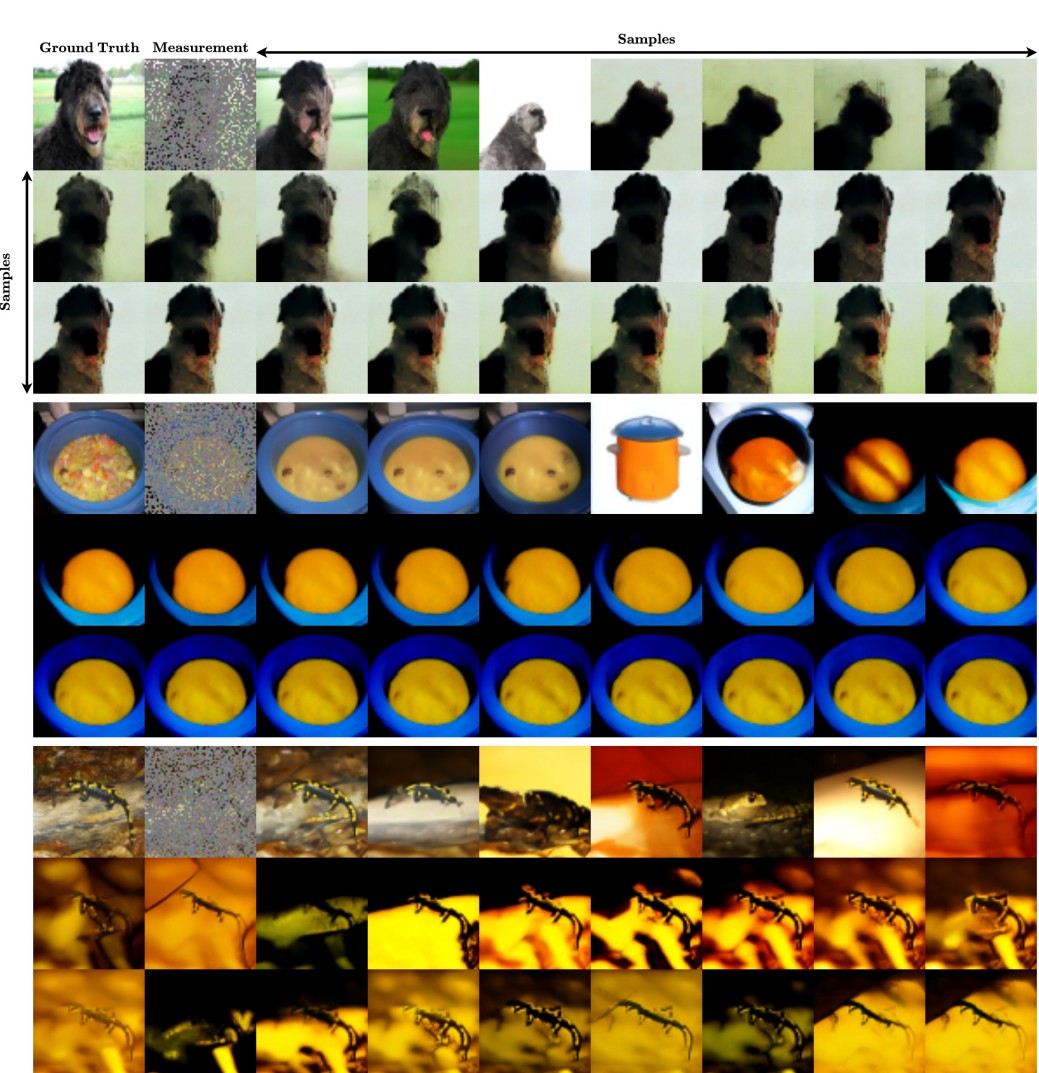

Figure A.12: Sets of samples for 20% inpainting on ImageNet (64 x 64).

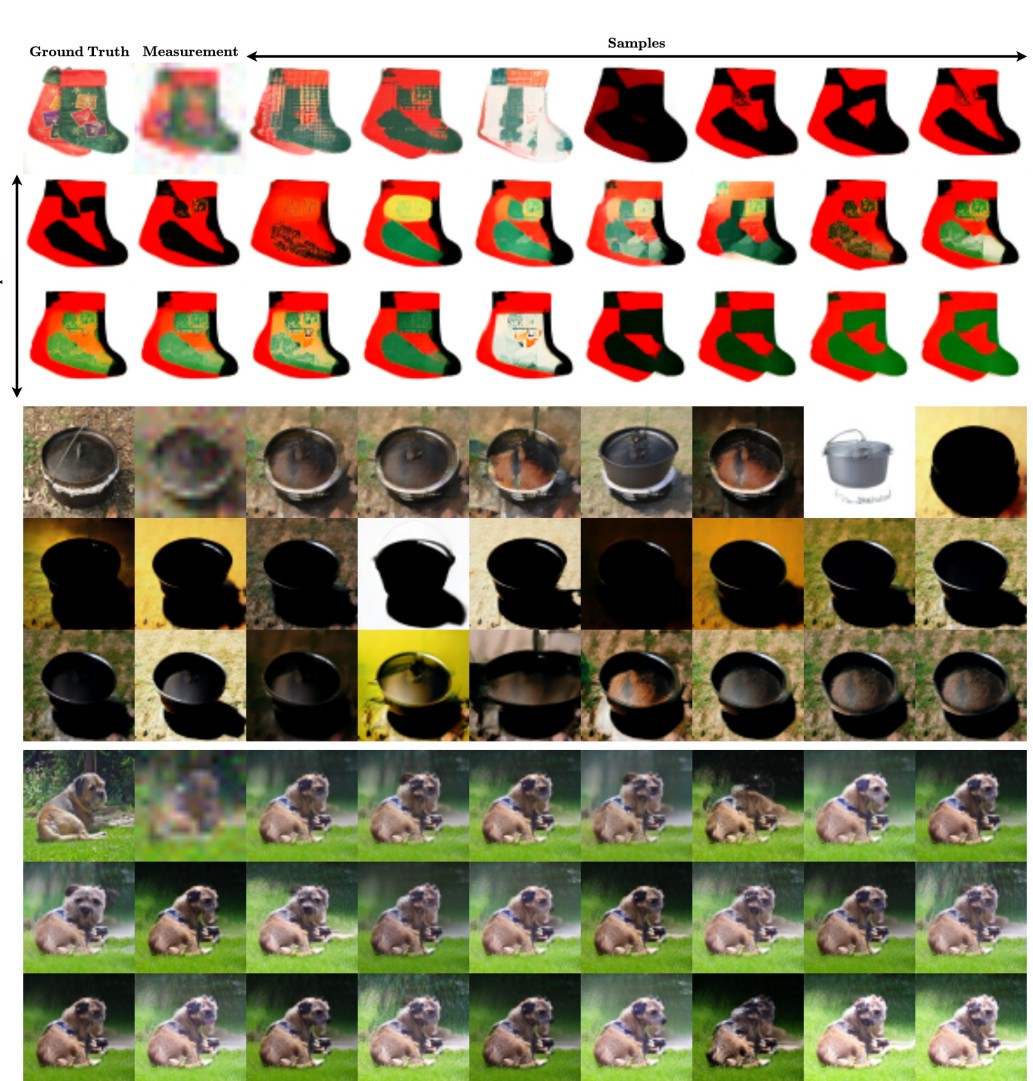

Figure A.13: Sets of samples for 4x super-resolution on ImageNet (64 x 64).

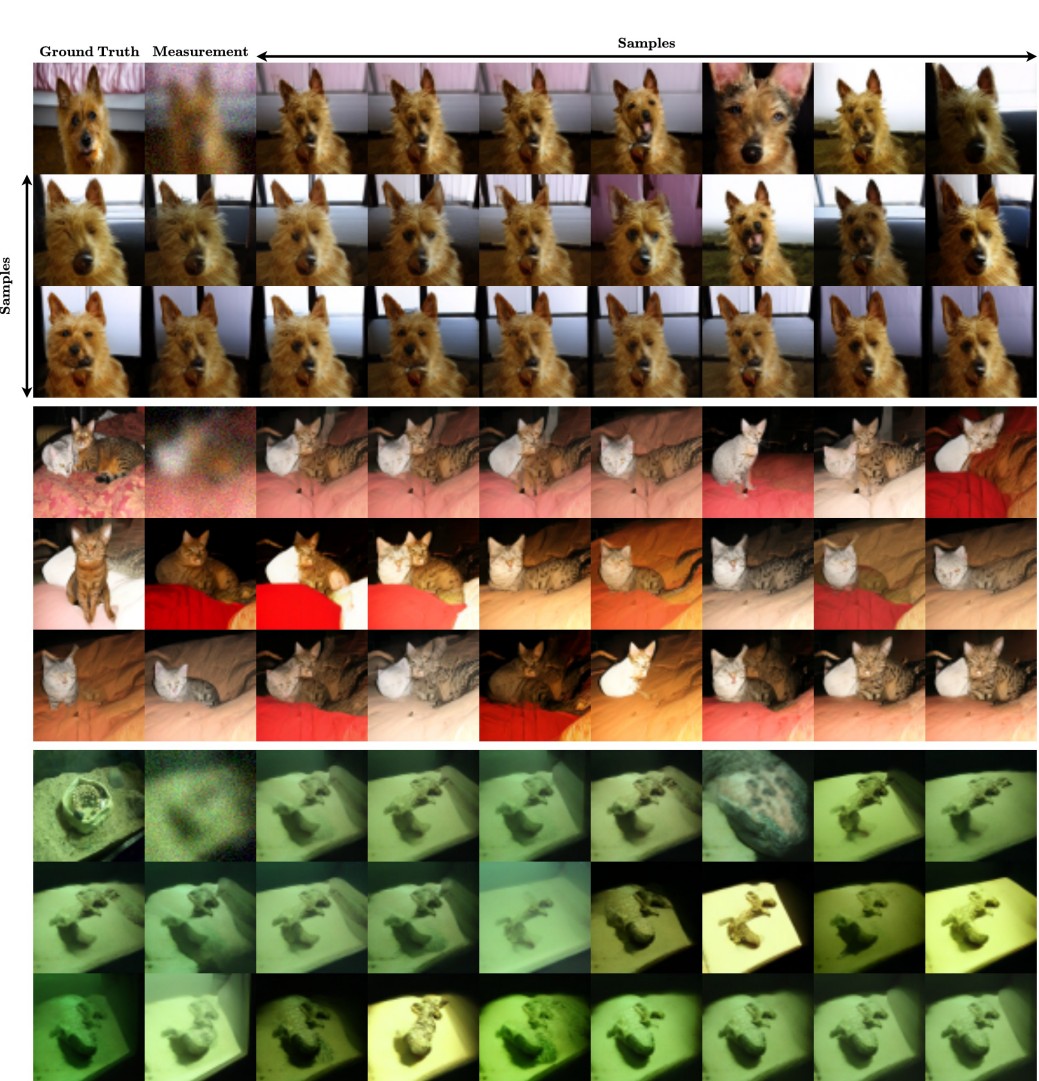

Figure A.14: Sets of samples for Gaussian deblurring on ImageNet (64 x 64).

