# OpenReview forum: "Posterior sampling via Langevin dynamics based on generative priors"
_ICLR.cc/2025/Conference — ICLR 2025 Conference Withdrawn Submission_

### Official Review · Reviewer_4Ft7 · 2024-10-23

**Soundness:** 2
**Presentation:** 2
**Contribution:** 2
**Rating:** 3
**Confidence:** 4

**Summary:**

This paper proposes a posterior sampling scheme for solving inverse problems in the latent space of consistency models.

**Strengths:**

**Originality** : to the best of my knowledge, this paper is the first to perform posterior sampling in the latent space of consistency models.

**Quality** : the proposed method is theoretically sound, and claims for diverse posterior sampling is supported with experiments on ImageNet 64x64 and LSUN bedroom 256x256.

**Clarity** : the proposed method and main results are clearly presented. I had no problem following the exposition.

**Significance** : while current diffusion-based image restoration methods often require large number of function evaluations, this paper introduces some ideas for fast image restoration.

**Weaknesses:**

**Limited Originality** : running MCMC in the latent space of one-step pushforward generative models is not new. Similar ideas are already explored in works such as [1,2] (none of which are cited in this paper). I also feel the paper is limited in its technical novelty, as it does not provide any insight into efficient posterior sampling in the latent space of more challenging multi-step priors such as diffusion models or hierarchical variational autoencoders.

**Overstatements** : I feel that the paper overstates the extent of its contributions. First of all, the proposed method *is not applicable to all types of generative priors*, but is compatible only with implicit generative models / pushforward generative models. Second, the proposed algorithm is, practically speaking, *not applicable to diffusion priors*, as it would require backpropagation through up to thousands to neural net compositions. The authors do mention adjoint sensitivity methods for backprop through diffusion in Section 5, but experiment results are not presented. Perhaps it is because it is too computationally expensive to run multiple steps of Langevin MCMC until mixing occurs.

**Missing Ablations** : the paper is missing ablations w.r.t. design choices in Section 5. For instance, how does the performance vary w.r.t. Euler Maruyama step size $\tau$? Is optimal $\tau$ consistent across dataset and data dimension?

[1] Neutra-lizing Bad Geometry in Hamiltonian Monte Carlo using Neural Transport

[2] MCMC Should Mix: Learning Energy-based Model with Neural Transport Latent Space MCMC

**Questions:**

**Q1** : in Figure 2, how do the authors check if posterior samples are distinct? For instance, for DPS, do distinct posterior samples mean samples initialized from distinct prior noise?

**Q2** : what are the unconditional FIDs / Inception Scores of diffusion models and consistency models used in Section 6?

**Q3** : have the authors tried using other types of gradient-based MCMC such as Hamiltonian Monte Carlo (HMC)? HMC can mix faster than Langevin Monte Carlo, so perhaps it could be combined with adjoint method for diffusion prior based posterior sampling.

---

### Official Review · Reviewer_Crbm · 2024-11-05

**Soundness:** 3
**Presentation:** 3
**Contribution:** 2
**Rating:** 6
**Confidence:** 4

**Summary:**

This paper proposes to use Langevin dynamics defined in latent space to perform posterior sampling of diffusion models, where the likelihood is defined in the clean data space. To evaluate the energy function, this paper proposes to use consistency models for significantly reducing the sampling cost. This paper also proposes a theoretical guarantee with mild assumptions. The empirical results show the effectiveness of the proposed method.

**Strengths:**

- The paper is well-written and easy to follow. The idea is simple but effective.
- The leveraging of consistency models significantly reduces the sampling costs, which is better than the previous data-prediction diffusion models.
- The experiments are solid.

**Weaknesses:**

This paper doesn't have major weakness, though there are two minor aspects:

- The method is slightly lack of novelty since the combination of consistency models and Lagevin dynamics is a direct generation of previous method such as DPS.
- The proposed method can only tackle with simple likelihood functions such as inpainting / deblurring / super resolution. It is unclear whether the proposed method can do complicated posterior inference in traditional Bayesian inference settings.

**Questions:**

I don't have specific questions and overall I recognize the contributions of this paper while I feel the novelty is not significant. So I prefer a boarderline accept.

---

### Official Review · Reviewer_afj4 · 2024-11-06

**Soundness:** 1
**Presentation:** 3
**Contribution:** 2
**Rating:** 3
**Confidence:** 3

**Summary:**

This paper introduced a new diffusion solver for inverse problem by using posterior sampling in the noise space. The method is training-free, and, therefore, is more efficient compared to some other related training-based frameworks. The authors also provide theoretical guarantees for the approximation error in total variation distance; along with empirical benchmarks to demonstrate the improvements in realistic image restoration/inpainting/super-resolution task.

**Strengths:**

- Well-motivated problem, the authors did a good literature review that lists relevant works.
- The derviation of the framework is based on establishing theories of SMC/denoising diffusion models. A theoretical analysis is always welcomed.
- The method is relatively straight-forward and easy to implement, and work on both linear and non-linear inverse problem settings.

**Weaknesses:**

- Huge doubt about practical performance (inference time): while the authors report low NFE, the total runtime of the sampling framework is not reported. The backpropagation through the whole pretrained consistency model $\phi$ for each steps are costly in both memory and computational time. I think it will be fairer to compare total wall-time with other baselines instead of just listing the NFEs as stated in the paper
- Unclear about the advantages of the proposed methods vs. baselines used in the benchmark: I also disagree on the argument of fair comparison to replace diffusion backbone by consistency models (CM) backbone to DPS and LGD  make them as fair as current CM-based framework in this paper (written around line 370-374). If DPS-DM and LGD-DM make faster inference *and* better quality reconstructed images, is it necessary calling them unfair comparison?
- Is the method completely training free? One of the key trick to make this method works, IMO, is the warm-start of the initial noise $z^0$, detailed around line 294-300, and on Appendix B.1. This is also related to the above point: the authors should also take into account runtime of the warm-start step and report it as requested on the point above.
- Missing strong baseline on linear inverse problem: the authors should include FPS (Dou & Song 2024) to the linear inverse posterior sampling baselines.

**Questions:**

As stated in weaknesses.

---

### Official Review · Reviewer_HaVH · 2024-11-08

**Soundness:** 2
**Presentation:** 2
**Contribution:** 1
**Rating:** 3
**Confidence:** 4

**Summary:**

This paper has proposed an efficient posterior sampling by simulating Langevin dynamics with a pre-trained generative model.

**Strengths:**

This paper has proposed an efficient posterior sampling method that could avoid running the full sampling chain.

**Weaknesses:**

1. The proposed method of posterior sampling using Langevin dynamics has been extensively studied and applied within the context of energy-based models. As a result, this approach lacks sufficient novelty for this work, given the established body of research already dedicated to similar methodologies.
2. The proposed method of sampling by posterior sampling by Langevin dynamics has been early explored in many EBM works such as [1] - [4] for multiple kinds of tasks, such as image generation, translation and saliency prediction. While there is no discussion about these work either in background or experiments.

3. Efficient sampling based on molecular dynamics have been well studies in many literatures such as [5] - [7], while these study lack of discussion in the paper.
4. The experiment is limited to the image reconstruction tasks, which raises concerns about the method’s generalizability to other domains or applications. Without testing on a broader range of scenarios, it is difficult to assess the model’s robustness and adaptability to different types of data or tasks.

[1] Xie, Jianwen, et al. "A theory of generative convnet." International conference on machine learning. PMLR, 2016.

[2] Xie, Jianwen, et al. "Cooperative learning of energy-based model and latent variable model via MCMC teaching." Proceedings of the AAAI Conference on Artificial Intelligence. Vol. 32. No. 1. 2018.

[3] Zhang, Jing, et al. "Learning generative vision transformer with energy-based latent space for saliency prediction." Advances in Neural Information Processing Systems 34 (2021): 15448-15463.

[4] Zhao, Yang, and Changyou Chen. "Unpaired image-to-image translation via latent energy transport." Proceedings of the IEEE/CVF conference on computer vision and pattern recognition. 2021.

[5] Gao, Ruiqi, et al. "Learning energy-based models by diffusion recovery likelihood." arXiv preprint arXiv:2012.08125 (2020).

[6] Gao, Ruiqi, et al. "Flow contrastive estimation of energy-based models." Proceedings of the IEEE/CVF Conference on Computer Vision and Pattern Recognition. 2020.

[7] Zhu, Yaxuan, et al. "Learning energy-based models by cooperative diffusion recovery likelihood." arXiv preprint arXiv:2309.05153 (2023).

**Questions:**

1. What is the difference between the proposed method with an Energy-Based Model?
2. Except for image reconstruction, could the proposed model be applied in other task?

---

### Official Review · Reviewer_qpVx · 2024-11-08

**Soundness:** 3
**Presentation:** 3
**Contribution:** 2
**Rating:** 3
**Confidence:** 4

**Summary:**

The paper proposes a method to do posterior sampling of samples, given certain conditions or partial information of the samples. The method assume there is a deterministic mapping between Gaussian noise and data, parametrized by e.g. a consistency model or a flow-based model. The posterior sampling is projected to and happens in the noise space with Langevin dynamics, and then the noise is projected back to the data space. Empirical results show that the proposed method leads to diverse and high quality samples in solving linear and nonlinear inverse problems.

**Strengths:**

* The paper is well written and easy to follow.
* Projecting the posterior sampling to a noise space is a good idea, as the posterior distribution in the noise space is close to a single modal Gaussian distribution, that is more friendly to MCMC.
* Theoretical analysis has been provided for the proposed method.

**Weaknesses:**

* My main concern is that the paper fails to put itself to the current position in the literature. This type of projecting MCMC sampling to a more MCMC friendly noise space has been well established in [1], and has been later on adapted to generative modeling regime in e.g. [2, 3], with the deterministic mapping being a VAE or a flow-based model. The contribution of this paper, positioned in these literation, is that it adapted the sampling to a posterior distribution, and leverages a CM with fixed noise as the deterministic mapping. In that case, I think the novelty is limited.

* The paper claims that the accumulation of samples leads to diverse samples. This needs to be further justified by analyzing the convergence behavior of the sampling chains. How do you make sure the samples from adjacent sampling steps are not correlated to each other (this could be guaranteed by other baseline methods that always start the sampling from independently sampled noise).

* The method assume that the sampling from the CM model is with fixed noise, which is lack of justification, and might partially explain why the sampling quality is suboptimal.

* Why are 1-step CM results better than 2-step CM empirically in general?

* Empirical results are not convincing enough. E.g., some samples in the top right row of figure 1 look oversaturated, which might indicate the sampling chain is not stable or mixing.


[1] NeuTra-lizing Bad Geometry in Hamiltonian Monte Carlo Using Neural Transport

[2] VAEBM: A Symbiosis between Variational Autoencoders and Energy-based Models

[3] MCMC Should Mix: Learning Energy-Based Model with Neural Transport Latent Space MCMC

**Questions:**

* Can you show the inpainting mask of samples in Figure 1 and Figure 5 in the updated draft?

---

### Note · Authors · 2024-11-15

I have read and agree with the venue's withdrawal policy on behalf of myself and my co-authors.